# Utility and precision evidence of technology in the treatment of type 1 diabetes: a systematic review

Laura M. Jacobsen[1,197], Jennifer L. Sherr[2,197], Elizabeth Considine [2], Angela Chen[1], Sarah M. Peeling[1], Margo Hulsmans[3], Sara Charleer [3], Marzhan Urazbayeva[4], Mustafa Tosur [4,5], Selma Alamarie[6], Maria J. Redondo[4], Korey K. Hood[6], Peter A. Gottlieb[7], Pieter Gillard [3], Jessie J. Wong [5], Irl B. Hirsch[8], Richard E. Pratley[9], Lori M. Laffel[10,198], Chantal Mathieu[3,198 ✉] & ADA/EASD PMDI*

## Abstract

**Background** The greatest change in the treatment of people living with type 1 diabetes in the last decade has been the explosion of technology assisting in all aspects of diabetes therapy, from glucose monitoring to insulin delivery and decision making. As such, the aim of our systematic review was to assess the utility of these technologies as well as identify any precision medicine-directed findings to personalize care.

**Methods** Screening of 835 peer-reviewed articles was followed by systematic review of 70 of them (focusing on randomized trials and extension studies with ≥50 participants from the past 10 years).

**Results** We find that novel technologies, ranging from continuous glucose monitoring systems, insulin pumps and decision support tools to the most advanced hybrid closed loop systems, improve important measures like HbA1c, time in range, and glycemic variability, while reducing hypoglycemia risk. Several studies included person-reported outcomes, allowing assessment of the burden or benefit of the technology in the lives of those with type 1 diabetes, demonstrating positive results or, at a minimum, no increase in self-care burden compared with standard care. Important limitations of the trials to date are their small size, the scarcity of pre-planned or powered analyses in sub-populations such as children, racial/ethnic minorities, people with advanced complications, and variations in baseline glycemic levels. In addition, confounders including education with device initiation, concomitant behavioral modifications, and frequent contact with the healthcare team are rarely described in enough detail to assess their impact.

**Conclusions** Our review highlights the potential of technology in the treatment of people living with type 1 diabetes and provides suggestions for optimization of outcomes and areas of further study for precision medicine-directed technology use in type 1 diabetes.

## Plain Language Summary

In the last decade, there have been significant advances in how technology is used in the treatment of people living with type 1 diabetes. These technologies primarily aim to help manage blood sugar levels. Here, we reviewed research published over the last decade to evaluate the impact of such technologies on type 1 diabetes treatment. We find that various types of novel technologies, such as devices to monitor blood sugar levels continuously or deliver insulin, improve important diabetes-related measures and can reduce the risk of having low blood sugar levels. Importantly, several studies showed a positive impact of technologies on quality of life in people living with diabetes. Our findings highlight the benefits of novel technologies in the treatment of type 1 diabetes and identify areas for further research to optimize and personalize diabetes care.

[1] University of Florida, Gainesville, FL, USA. [2] Yale School of Medicine, New Haven, CT, USA. [3] University Hospital Leuven, Leuven, Belgium. [4] Baylor College of Medicine, Texas Children's Hospital, Houston, TX, USA. [5] Children's Nutrition Research Center, USDA/ARS, Houston, TX, USA. [6] Stanford University School of Medicine, Stanford, CA, USA. [7] Barbara Davis Center, University of Colorado School of Medicine, Aurora, CO, USA. [8] University of Washington School of Medicine, Seattle, WA, USA. [9] AdventHealth Translational Research Institute, Orlando, FL, USA. [10] Joslin Diabetes Center, Harvard Medical School, Boston, MA, USA. [197] These authors contributed equally: Laura M. Jacobsen, Jennifer L. Sherr. [198] These authors jointly supervised this work: Lori M. Laffel, Chantal Mathieu. *A list of authors and their affiliations appears at the end of the paper. ✉ email: chantal.mathieu@uzleuven.be

At the time of the centennial anniversary of the first clinical use of insulin, the treatment of type 1 diabetes has undergone multiple innovations that have advanced the health, well-being, and longevity of people living with the disease[1]. There have been numerous improvements in insulin formulations, in particular, the generation of insulin analogs to create ultra-rapid prandial onset or prolonged basal insulin action aiming at more closely mimicking normal physiology. While these newer insulin preparations helped more people with type 1 diabetes achieve targeted glucose levels, further glycemic improvements required advancements in glucose monitoring technologies. Most recently, factory-calibrated continuous glucose monitors (CGMs) have all but eliminated the need for self-monitoring of blood glucose (SMBG)[2–4].

The discovery and routine availability of hemoglobin A1c (HbA1c) measurements in the 1970s provided for the quantification of overall glycemia, which, in turn, supported the design and implementation of the Diabetes Control and Complications Trial (DCCT)[5]. This landmark study compared intensive insulin therapy, consisting of multiple daily insulin injections (MDI) or continuous subcutaneous insulin infusion (CSII) pump therapy with SMBG, with conventional insulin therapy, consisting of only one to two daily injections of insulin with urine glucose monitoring. The DCCT confirmed the importance of intensive insulin therapy versus conventional insulin therapy in improving glycemic levels and reducing the risk of diabetic nephropathy, retinopathy, and neuropathy, as well as long-term adverse cardiovascular outcomes[5–7]. Since 1993, intensive insulin therapy has remained the mainstay of treatment of type 1 diabetes, albeit with a substantial self-care burden placed upon the person living with type 1 diabetes until the creation of advanced technologies that have eased many self-care demands.

This systematic review focuses on these advanced diabetes technologies for the treatment of type 1 diabetes as a part of the second International Consensus Report of the Precision Medicine in Diabetes Initiative (PMDI)[8]. The PMDI was established in 2018 by the American Diabetes Association (ADA) in partnership with the European Association for the Study of Diabetes (EASD). The ADA/EASD PMDI includes global thought leaders in precision diabetes medicine who are working to address the burgeoning need for better diabetes prevention and care through precision medicine[9].

We recognize that type 1 diabetes treatment requires orchestrated education and support around a myriad of activities, including dietary intake, exercise management, insulin administration, glucose monitoring, adjunctive therapies, and behavioral health, along with transitions in care across the lifespan, all of which should be tailored to the individual's needs with a precision medicine approach. Given the rapid evolution of technologies, we have limited our systematic review to the last 10 years of published research on advanced diabetes technologies used in the treatment of type 1 diabetes. While regulatory bodies have recognized glycemic control measured as HbA1c as a proximate outcome of various type 1 diabetes treatments, over the past decade, glycemic outcomes have evolved to include assessment of glucose time in range (TIR) and glucose time below range (TBR) as well as other glucometrics associated with CGM use[10,11]. Therefore, we include multiple outcomes in our assessments of technologies for the treatment of type 1 diabetes, including those related to person-reported outcomes (PROs), given the opportunity for technology to mitigate self-care burden. We have sought to evaluate the available evidence to answer the questions as to the (1) efficacy of each technology along with the quality of the evidence, (2) whether individual characteristics can help identify which persons living with type 1 diabetes are likely to derive these therapeutic benefits from the available advanced diabetes technologies, and (3) when in the lifecycle should such approaches be implemented to optimize glycemia, reduce severe hypoglycemia, and preserve health for those living with type 1 diabetes. Through our systematic approach, we find fairly strong evidence for the use of all technologies in the treatment of type 1 diabetes and early data to support a precision approach to implementation using an individual's personal goals as well as age.

## Methods

**Protocol development**. Charged with assessing treatment modalities in people with type 1 diabetes, a group of 13 experts identified key areas of type 1 diabetes management. A protocol for this review was developed and registered on 11/17/2021 with PROSPERO (CRD42021271680) before implementation. Initially, nine areas of interest were identified with topics including adjunctive to insulin therapies, behavioral health, beta cell replacement, exercise, glycemic targets, intensive insulin therapy, nutrition, transition of care, and technology for the management of diabetes (Supplementary Table 1). Recognizing that the biggest change over the past decade has been in technological advances for type 1 diabetes management, our protocol was amended (2/2/2023) to focus solely on this topic.

**Search strategy, study selection, and screening process**. Using PubMed and EMBASE, Medical Subject Headings (MeSH) and free-text terms (Supplementary Table 2) were used to identify technology-related studies. To assess the sensitivity of the search terms, five key articles were chosen independently, and each was identified through the search.

Abstracts retrieved from the literature search were loaded into the Covidence platform. Eligible studies included individuals with type 1 diabetes or Latent Autoimmune Diabetes in Adults (LADA), randomized controlled trials (RCTs), with a minimum of 50 participants, full text available in English, and published January 1st, 2012 through September 5th, 2022. Secondary papers, extension studies, and PRO-focused studies stemming from RCTs were also included. Studies were only included if they occurred in ambulatory settings. Full inclusion and exclusion criteria are listed in Supplementary Table 3. Given the rapid evolution of diabetes technologies, the period was restricted to approximate the past decade. Recognizing that smaller studies may be prone to either type I or type II errors, the group came to a consensus regarding the need for a minimum sample size of 50 participants in order to include a trial. The primary goal of the review was to determine whether characteristics of people living with type 1 diabetes across the lifespan could identify the best (tailored) treatment(s) to optimize outcomes, including HbA1c, TIR, weight, hypoglycemia, quality of life, and other measures. A priori in the Prospero protocol, subgroups to be considered for analysis were determined and are included in Supplementary Table 4.

**Data extraction**. At each stage, a minimum of two authors screened titles and abstracts found in the literature search. Disagreements about inclusion were resolved by consensus after consultation with a third reviewer. The next level of screening included reviewing the full text of each publication identified in abstract screening.

Covidence software was used to extract data from eligible full texts by two team members in tandem. A standardized template was made in Covidence to aid with data extraction from each full-text article to document journal citation information, study design, and population, limitations, comparison group/controls, intervention, outcomes including subgroup analyses, and limitations.

**Quality assessment extraction.** Cochrane risk of bias (https://methods.cochrane.org/bias/resources/rob-2-revised-cochrane-risk-bias-tool-randomized-trials) was used to assess the quality of the studies as the identified texts were limited to RCTs. Therefore, two team members assessed sequence generation, allocation concealment, blinding of participants/personnel, blinding of outcome assessment, incomplete outcome data, selective reporting, and other sources of bias to determine the overall risk of bias. For the purposes of this quality assessment, other sources of bias were deemed present if multiple comparisons were done without appropriate correction, which would lead to a high risk of bias, especially in subgroup analyses. If the information required to assess quality for each domain was not found in the manuscript, it was coded as not reported. GRADE criteria were used to rate the quality of evidence for a given topic[12].

**Data analysis and synthesis.** Due to the heterogeneity of technologies used and outcome measures reported, it was not feasible to perform a meta-analysis. Instead, a narrative summary of findings by technology type is presented here.

## Results

A total of 835 studies were screened, of which 70 citations met pre-specified parameters to be included in this report (Fig. 1). Of the trials identified, 45 were primary reports of RCT data, 6 reviewed data from an extension phase of an RCT, and 10 were secondary outcome papers. (Fig. 2). Studies describing PROs accounted for 9 manuscripts (Fig. 2). Quality assessment is visualized in Fig. 3 for each of these types of analyses. While the trials were balanced regarding the proportion of females and males, the populations studied were overwhelmingly non-Hispanic white. Of note, over the last decade, the technologies studied have shifted towards higher levels of automation (Fig. 4).

**Continuous glucose monitoring.** Technologies allowing continuous measurement of ambient glucose levels in subcutaneous tissue have gained rapid traction in people living with type 1 diabetes. These technologies not only allow insight into glycemic levels without invasive SMBG but provide continuous data with trend analysis and (depending on the systems) alerts of impending hyper- or hypoglycemia. Initially used as diagnostic tools to assist diabetes care teams, they have rapidly become, at least in medium- and high-income countries, the standard of care in glucose measurement for people living with type 1 diabetes. Even prior to regulatory approval, these devices have assisted many people with diabetes in their day-to-day decision-making on insulin dosing and provide the foundations for advanced diabetes technologies that tie insulin dosing to sensor glucose levels.

In recent years, many different technologies have become available, with rapid evolution in systems (type of sensor, type of reporting, compatibility of the system with smartphones, integration of reporting in EMR, affordability)[4]. At present, two types of CGM systems are used, depending on the level of user interaction needed to receive sensor glucose information. Real-time CGM (rtCGM) provides updated glycemic information every 1–5 min through a continuous connection between the transmitter and receiver, offering real-time alerts. Intermittently scanned CGM (isCGM), also known as flash glucose monitoring, provides the same glycemic information as rtCGM but requires the user to deliberately scan the sensor at least every eight hours to obtain complete glucose data.

As shown in Table 1, the literature search identified 27 articles, 15 primary analyses[13–27], 4 secondary analyses[28–31], 4 extension studies[32–35], and 4 PROs[36–39]. Of primary analyses, 13 were RCTs using parallel ($n=10$) or crossover ($n=3$) design comparing CGM to SMBG. The majority of studies investigated the standalone use of rtCGM[14,16–19,21,22,28,31,32,34,35], while some also examined the use of diagnostic ('professional') rtCGM[15], do-it-yourself rtCGM[24], standalone isCGM[20,23,29,30], and decision support systems on top of CGM[13,23]. The number of included participants ranged from 52 to 448, with a wide age range (from children to older adults). Three studies had a duration of more than 12 months[28,33,38], with most studies having a maximum of 6 months of follow-up[13–26,29–32,34,35,37–39]. Eleven of the trials (40%) were sponsored by industry, with 5 of the remaining trials reporting in-kind support of devices.

Most RCTs (primary analysis) had HbA1c as the primary outcome[13,15,17,19,21,22,25,26,33,34], with hypoglycemia as the primary outcome in three studies[16,18,20] and TIR[14,23,32] in two studies. One study included the Parental Hypoglycemia Fear Survey as the primary outcome[24]. Reported secondary outcomes

```
┌─────────────────────────┐      ┌ ─ ─ ─ ─ ─ ─ ─ ─ ─ ─ ─ ─ ┐
│ 835 Studies Imported    │─────▶  22 Duplicates removed
│ for Screening           │      └ ─ ─ ─ ─ ─ ─ ─ ─ ─ ─ ─ ─ ┘
└─────────────────────────┘
           │
           ▼
┌─────────────────────────┐      ┌ ─ ─ ─ ─ ─ ─ ─ ─ ─ ─ ─ ─ ┐
│ 813 Studies Screened    │─────▶  613 studies irrelevant
└─────────────────────────┘      └ ─ ─ ─ ─ ─ ─ ─ ─ ─ ─ ─ ─ ┘
           │
           ▼
┌─────────────────────────┐      ┌ ─ ─ ─ ─ ─ ─ ─ ─ ─ ─ ─ ─ ┐
│ 200 studies sought for  │─────▶  29 studies had no full text
│ retrieval               │      └ ─ ─ ─ ─ ─ ─ ─ ─ ─ ─ ─ ─ ┘
└─────────────────────────┘
           │
           ▼
┌─────────────────────────┐      ┌ ─ ─ ─ ─ ─ ─ ─ ─ ─ ─ ─ ─ ┐
│ 171 full-text studies   │─────▶  101 studies excluded
│ assessed for eligibility        Wrong comparator (n=1)
│ studies sought for      │        Wrong indication (n=2)
└─────────────────────────┘        Wrong intervention (n=1)
           │                       Wrong study design (n=97)
           ▼                     └ ─ ─ ─ ─ ─ ─ ─ ─ ─ ─ ─ ─ ┘
┌─────────────────────────┐
│ 70 studies included     │
└─────────────────────────┘
```

**Fig. 1 PRISMA flow chart of all included studies.** Of 835 studies, 70 were included in the final review.

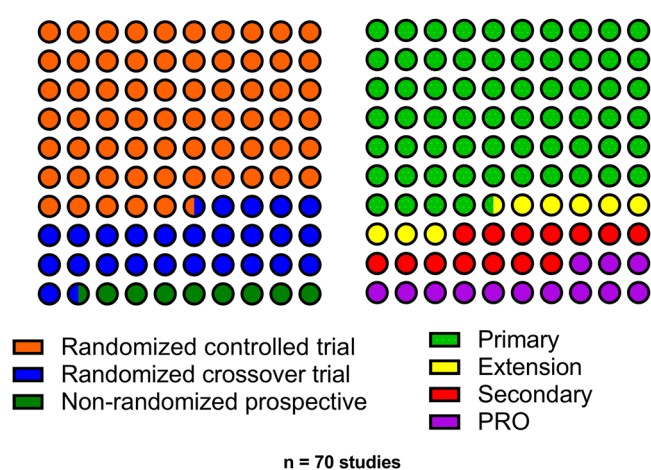

**n = 70 studies**

Legend:
- 🟧 Randomized controlled trial
- 🟦 Randomized crossover trial
- 🟩 Non-randomized prospective
- 🟩 Primary
- 🟨 Extension
- 🟥 Secondary
- 🟪 PRO

**Fig. 2 Distribution of study types included in the systematic review.** The proportion of studies ($n=70$) included in the systematic review by clinical trial design and analysis type (primary trial results, extension study of a randomized controlled trial [RCT], secondary analysis of an RCT, or person-reported outcome [PRO] studies).

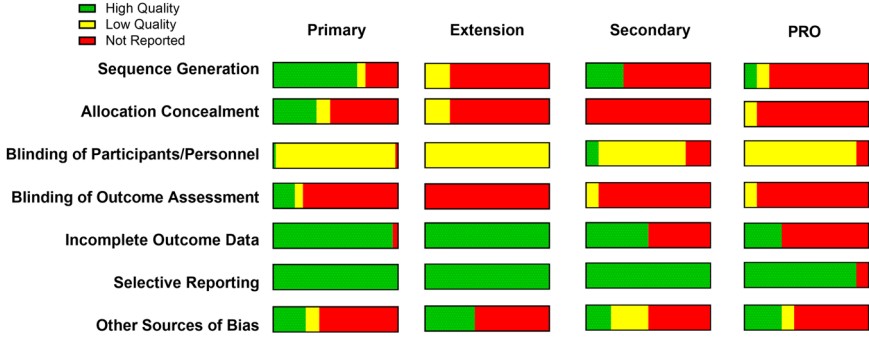

**Fig. 3 The proportion of studies with a low risk of bias (high quality) or high risk of bias (low quality).** This quality assessment is visualized by each individual quality criteria and grouped by the analysis type (primary trial results, extension studies, secondary analyses, or person-reported outcome [PRO] studies). Other sources of bias refer to the presence or absence of a correction for multiple testing.

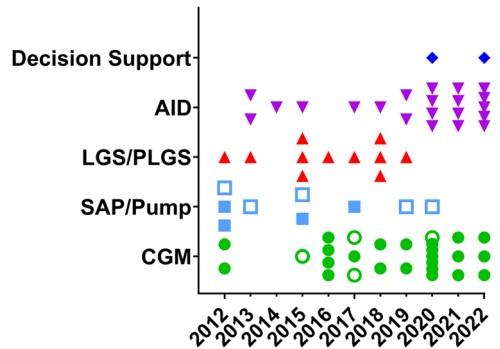

**Fig. 4 The type of technology studied within each published trial is plotted over time by the year of publication.** Each symbol is an individual publication using the specified technology listed: decision support, automated insulin delivery (AID), low-glucose suspend/predictive low-glucose suspend (LGS/PLGS), sensor-augmented pump (SAP)/pump, and continuous glucose monitoring (CGM). Open symbols are person-reported outcome (PRO) studies.

were diverse, encompassing many aspects of diabetes management and well-being (Table 1).

When comparing the method of glucose monitoring (CGM vs. SMBG), half of the RCTs found no statistically significant difference in mean HbA1c[14,18–20,25,29,32], while the other half showed a benefit on HbA1c (mean between-group difference at study end 0.23–0.60% in favor of CGM)[16,17,21,22,31,34]. The most consistent finding was the superior effect of CGM on hypoglycemia prevention and improvement in TIR, which was demonstrated in all but two studies in children[19,32].

Of the articles included, 15/27 had a low risk of bias for sequence generation, and 7/27 had a low risk of bias for allocation concealment. Both study personnel and participants were aware of the treatment group in all of the studies. Blinding of outcome assessment could not be assessed in all but 6 papers. The vast majority had complete data and minimal bias for selective reporting. The writing group acknowledges several limitations of the studies evaluating CGM in people living with type 1 diabetes, the short duration of observation, open design of the study, lack of diversity of the populations studied, and the speed of the evolution of technologies (some obsolete by the time of publication). (GRADE evidence: Level B).

Despite a rigorous literature search, some articles fulfilling the criteria were not identified. For example, the ALERT1 study prospectively compared isCGM without alerts with rtCGM with alerts in adults living with type 1 diabetes treated with MDI[40].

This 6-month study included 254 randomized participants and demonstrated superiority in achievement in TIR, HbA1c, and hypoglycemia risk, including severe hypoglycemia[40]. Some primary trial results were not identified, like the Strategies to Enhance New CGM Use in Childhood (SENCE) which was a study of 143 children aged 2 to <8 years who were randomized to the use of CGM alone, CGM with a familial behavioral intervention, and SMBG that showed no change in TIR but reduced time in hypoglycemia[41]; yet, an extension of the cohort led by Van Name was included[32]. A similar issue occurred with the GOLD trial, which assessed CGM use in 161 adults with type 1 diabetes on MDI, where the primary study was not included[42]. Further, publications after our literature search on 9/5/2022 were not included. Focusing on the ALERT1 study, since then, the group has published two additional papers. One described an 18-month partial cross-over extension where continued benefits of rtCGM with alarms were demonstrated[43], while the other showed that subgroup analysis could not identify any predictive factor differentiating glycemic benefit in people with type 1 diabetes[44].

**Decision support.** Despite advances in diabetes technologies, for many living with type 1 diabetes, achievement of glycemic targets remains elusive[45]. Some of this may be due to an insufficient number of providers who can guide diabetes management and the frequency with which follow-up is often scheduled. To address this gap and streamline the process of integrating data to optimize insulin doses, decision support systems have been developed in recent years. We identified only two relevant articles reporting primary analysis results of multicenter RCTs on decision support systems (Table 2)[46,47]. One of the trials was industry-sponsored. These studies randomized 80 and 122 individuals, respectively, and included both adult and pediatric participants. Bisio et al. studied a CGM-based decision support system, including real-time dosing advice and retrospective therapy optimization in individuals with type 1 diabetes using MDI over 3 months[46]. Nimri et al. evaluated an automated artificial intelligence-based decision support system in participants with type 1 diabetes using insulin pump therapy and CGM over 6 months[47]. The primary outcome was CGM-based TIR for both studies. While Bisio et al. failed to show a significant difference in TIR (or HbA1c, analyzed as a secondary outcome)[46], Nimri et al. demonstrated that TIR in those using the decision support system was not inferior to intensive insulin titration provided by physicians[47]. Although Nimri et al. also reported statistically significant improvement in HbA1c from baseline to the end of the study in those randomized to the decision support system, there was no between-group difference in HbA1c between the study arms[47]. Information on sequence generation, allocation

**Table 1 CGM-related clinical trial publications included in the systematic review (2012–2022).**

| Author, year of publication; Trial Cohort if applicable; analysis type: primary (P), secondary (S), extension (E), (PRO) | System or product studied | Locations (trial sites) | Study design[a] | Age range (yrs) | HbA1c (%) range | # Randomized | Trial duration (mos) | Primary outcome | Comparator/assessed measure | Sig (Y/N) | Funding |
|---|---|---|---|---|---|---|---|---|---|---|---|
| Mauras[25] DirecNet (P) | Abbott FreeStyle Flash | USA | RCT | 4–9 | ≥7 | 146 | 6 | HbA1c | SMBG | N | NIH-NICHHD, -NCRR, -RMR |
| Battelino[22] SWITCH (P) | Medtronic Guardian REAL-Time Clinical | Austria, Denmark, Italy, Luxembourg, Netherlands, Slovenia, Spain | Rcrossover | 6–70 | 7.5–9.5 | 153 | 6 | HbA1c | SMBG | Y | Medtronic |
| Rasbach[38] (PRO) | Dexcom[b] | USA | RCT | 8–17 | 6.5–10 | 120 | 24 | Survey | CGM self-efficacy | Y | NIH-NIDDK, Katherine Adler Astrove Youth Education Fund, Maria Griffin Drury Pediatric Fund, Eleanor Chesterman Beatson Fund |
| Bolinder[20] IMPACT (P) | Abbott FreeStyle Libre flash[b] | Austria, Germany, Netherlands, Spain, Sweden | RCT | 2–17 | 7.5–11 | 241 | 6 | Hypoglycemia | SMBG | Y | Abbott |
| El-Laboudi[31] JDRF-CGM (S) | Dexcom Seven, Medtronic Enlite, or Abbott Freestyle Libre[b] | UK | RCT | ≥8 | 7–10 | 448 | 6 | Glycemic variability | SMBG | Y | |
| vanBeers[14] IN CONTROL(P) | Medtronic Enlite | Netherlands | Rcrossover | 18–70 | Any | 52 | 4 | TIR | SMBG | Y | Eli Lilly and Sanofi, in-kind support Medtronic |
| Soupal[26] COMISAIR (P) | Medtronic Enlite or Dexcom G4 | Czech Republic | NRP | >18 | 7–10 | 65 | 12 | HbA1c | SMBG | Y | Agency for Healthcare Research (AZV) Czech Rep. |
| Polonsky[39] DIaMonD (PRO) | Dexcom G4 | USA | RCT | ≥25 | 7.5–10 | 158 | 6 | Survey | QOL | Y | Dexcom |
| vanBeers[37] IN CONTROL (PRO) | Medtronic Enlite | Netherlands | Rcrossover | 18–70 | Any | 52 | 11 | Survey | Emotional well-being | Y | Eli Lilly and Sanofi |
| Beck[21] DIaMonD (P) | Dexcom G4 | USA | RCT (2:1) | ≥25 | 7.5–10 | 158 | 6 | HbA1c | SMBG | N | Dexcom |
| Heinemann[18] HypoDE (P) | Dexcom G5 | Germany | RCT | ≥18 | ≤9 | 163 | 6 | Hypoglycemia | SMBG | N | Dexcom |
| Oskarsson[29] IMPACT (S) | Abbott FreeStyle Libre flash[b] | Austria, Germany, Netherlands, Spain, Sweden | RCT | ≥18 | ≤7.5 | 167 | 6 | TBR | SMBG | N | Abbott |
| Guilmin-Crepon[19] Start-In! (P) | Medtronic Enlite | France | RCT (1:1:1) | 2–17 | 7.5–11 | 151 | 12 | HbA1c | SMBG | N | French Health Ministry |
| Raviteja[15] (P) | Medtronic iPRO 2 Prof. CGM | India | RCT | 2–10 | <12 | 68 | 3 | HbA1c | SMBG | Y | |
| Marsters[30] (S) | Abbott FreeStyle Libre flash[b] | New Zealand | RCT | 13–20 | ≥9 | 64 | 6 | Cutaneous | SMBG | Y | Cure Kids, Australasian Pediatric Endocrine Group, University of Otago |

**Table 1 (continued)**

| Author; year of publication; Trial Cohort if applicable; analysis type: primary (P), secondary (S), extension (E), (PRO) | System or product studied | Locations (trial sites) | Study design[a] | Age range (yrs) | HbA1c (%) range | # Randomized | Trial duration (mos) | Primary outcome | Comparator/ assessed measure | Sig (Y/N) | Funding |
|---|---|---|---|---|---|---|---|---|---|---|---|
| Pratley[16] WISDM (P) | Dexcom G5 | USA | RCT | ≥60 | <10 | 203 | 6 | TBR | SMBG | Y | JDRF, Leona M. and Harry B. Helmsley Charitable Trust (HCT), NIH-NCRR, -NCATS, in-kind support Dexcom |
| Soupal[33] COMISAIR (E) | Medtronic Enlite or Dexcom G4 | Czech Republic | NRP | >18 | 7–10 | 94 | 36 | HbA1c | SMBG | Y | AZV, Ministry of Health Czech Rep, Dexcom |
| AlHayek[27], April (P) | Abbott FreeStyle Libre flash[b] | Saudi Arabia | NRP | 13–19 | – | 187 | 3 | Survey | Diabetes distress | Y | None for study conduct; Publication fees by Abbott |
| AlHayek[36], July (PRO) | Abbott FreeStyle Libre flash[b] | Saudi Arabia | NRP | 18–40 | – | 95 | 3 | Survey | Diabetes distress | Y | None for study conduct; Publication fees by Abbott |
| Seyed-Ahmadi[28] GOLD (S) | Dexcom G4 | Sweden | Rcrossover | ≥18 | ≥7.5 | 137 | 17.25 | TBR | SMBG | Y | Swedish State (ALF agreement) HCT, in-kind support Dexcom |
| Laffel[17] CITY (P) | Dexcom G5 | USA | RCT | 14–24 | 7.5–11 | 153 | 6 | HbA1c | SMBG | Y | |
| Lind[34] SILVER (E) | Dexcom G4 | Sweden | Rcrossover | ≥18 | ≥7.5 | 107 | 12 | HbA1c | SMBG | Y | Swedish State (ALF), Dexcom |
| Secher[23] (P) | Abbott FreeStyle Libre flash[b] | Denmark | RCT (1:1:1:1) | ≥18 | >7 | 170 | 6 | TIR | SMBG | N | Capital Region of Denmark |
| Xu[13] (P) | Abbott FreeStyle Libre 1 flash | China | RCT (1:1:1) | 10–19 | 7–10 | 80 | 6 | HbA1c | SMBG | Y | National Key Research and Development Program, NSF of China, Shanghai Municipal Education Commission-Gaofeng Clinical Medicine |
| Elbalshy[24] (P) | Abbott FreeStyle Libre flash vs. addition of MiaoMiao version 2 | New Zealand | Rcrossover | 2–13 | – | 55 | 4 | Survey | CGM, Parental FOH | N | Dept Women's & Children's Health Research Grant, Dunedin School of Medicine, Freemasons New Zealand, Health Research Council, Lottery Health Research |
| Miller[35] WISDM (E) | Dexcom G5 | USA | RCT | ≥60 | <10 | 203 | 12 | Hypoglycemia | SMBG | Y | JDRF, HCT, Public Health Service Research Grant, in-kind support Dexcom |
| VanName[32] SENCE (E) | Dexcom G5 | USA | RCT (1:1:1) | 2–8 | 7–9 | 143 | 12 | TIR | SMBG | N | HCT, in-kind support Dexcom |

[a] All trials, if randomized, were randomized 1:1 unless the randomization scheme is listed in this column. RCT randomized controlled trial, Rcrossover T randomized crossover trial, NRP non-randomized prospective.
[b] No further specification of the device in the paper.
SMBG self-monitoring blood glucose, TIR time in range (70–180 mg/dL), PRO patient-reported outcome, TBR time below range (<70 mg/dL).

**Table 2 Pump, SAP, LGS, PLGS, and decision support tools-related clinical trial publications included in the systematic review (2012-2022).**

| Author; year of publication; trial cohort if applicable; analysis type: primary (P), secondary (S), extension (E), (PRO) | System or product studied | Locations (trial sites) | Study design[a] | Age range (yrs) | HbA1c (%) Range | # Randomized | Trial duration (mos) | Primary outcome | Comparator/ assessed measure | Sig (Y/N) | Funding |
|---|---|---|---|---|---|---|---|---|---|---|---|
| *Pump/SAP* | | | | | | | | | | | |
| Kordonouri[57] Pediatric Onset Study (E) | Medtronic Paradigm, Enlite sensor | Austria, France, Germany, Poland | RCT | 1-16 | NA | 131 | 24 | HbA1c | SMBG/MDI | N | Medtronic |
| Slover[56] STAR 3 (S) | Medtronic Paradigm REAL-Time | USA, Canada | RCT | 7-18 | 7.4-9.5 | 156 | 12 | HbA1c | MDI | Y | Medtronic |
| Rubin[61] STAR 3 (PRO) | Medtronic Paradigm REAL-Time | USA, Canada | RCT | 7-70 | 7.4-9.5 | 485 | 12 | Survey | HRQOL | Y | Medtronic |
| Peyrot[60] STAR 3 (PRO) | Medtronic Paradigm REAL-Time | USA, Canada | RCT | 7-70 | 7.4-9.5 | 485 | 12 | Survey | Satisfaction | Y | Medtronic |
| Tanenberg[59] STAR 3 (PRO) | Medtronic Paradigm REAL-Time and Sof-Sensor | USA, Canada | RCT | 7-70 | 7.4-9.5 | 244 | 12 | HbA1c | SAP | Y | Medtronic, in-kind support LifeScan, Bayer Healthcare, Becton Dickinson |
| Rosenlund[55] (P) | Medtronic Paradigm Veo, Enlite sensor | Denmark | RCT | 18-75 | ≥7.5 | 60 | 12 | UACR | SMBG/MDI | N | Medtronic |
| Beck[54] DiaMonD (P) | Insulet Omnipod and Dexcom G4 | USA | RCT | ≥25 | 7.5-10 | 75 | 6.5 | TIR | MDI | Y | Dexcom |
| Speight[58] HypoCOMPaSS (PRO) | Medtronic Paradigm Veo System | UK | RCT | 18-74 | - | 96 | 24 | Survey | Satisfaction | Y | Diabetes UK |
| Messe[62] (PRO) | NA | USA | NRP | 12-19 | - | 411 | 0 | Survey | QOL | Y | HCT |
| *LGS/PLGS* | | | | | | | | | | | |
| Bergenstal[66] ASPIRE (P) | Medtronic Paradigm Veo, Enlite sensor | USA | RCT | 16-70 | 5.8-10 | 247 | 3 | Hypoglycemia | SAP | Y | Medtronic |
| Ly[72] (P) | Medtronic Paradigm Veo, Enlite sensor | Australia | RCT | 4-50 | ≤8.5 | 95 | 6 | Hypoglycemia | SAP | Y | JDRF, in-kind support Medtronic |
| Buckingham[68] IN HOME CLOSED LOOP (P) | Medtronic Paradigm Veo, Enlite sensor | USA, Canada | Rcrossover | 4-14 | ≤8.5 | 81 | 1.3 | TBR | SAP | Y | NIH-NIDDK, JDRF, JDRF CCTN |
| Weiss[73] May ASPIRE (S) | Medtronic Paradigm Veo System | USA | RCT | 16-70 | 5.8-10 | 247 | 3 | Hypoglycemia | NH predictors | Y | Medtronic |
| Weiss[74] Aug ASPIRE (S) | Medtronic Paradigm Veo System | USA | RCT | 16-70 | 5.8-10 | 247 | 3 | Hypoglycemia | Correlate A1c/NH | Y | Medtronic |

**Table 2 (continued)**

| Author; year of publication; trial cohort if applicable; analysis type: primary (P), secondary (S), extension (E), (PRO) | System or product studied | Locations (trial sites) | Study design[a] | Age range (yrs) | HbA1c (%) Range | # Randomized | Trial duration (mos) | Primary outcome | Comparator/ assessed measure | Sig (Y/N) | Funding |
|---|---|---|---|---|---|---|---|---|---|---|---|
| Calhoun[69] IN HOME CLOSED LOOP (S) | Medtronic Paradigm Veo, Enlite sensor | USA, Canada | Rcrossover | 4–45 | ≤8.5 or ≤8.0[b] | 127 | 1.5 | Hypoglycemia | SAP | Y | NIH-NIDDK, JDRF, JDRF CCTN |
| Battelino[65] (P) | Medtronic 640 G and Enlite sensor | Israel, Slovenia | RCT | 8–18 | ≤10 | 100 | 0.5 | Hypoglycemia | SAP | Y | Faculty of Medicine, University of Ljubljana, Slovene National Research Agency, in-kind support Medtronic |
| Forlenza[70] PROLOG (P) | Tandem t:slim X2 with Basal-IQ and Dexcom G5 | USA | Rcrossover | ≥6 | – | 103 | 1.5 | TBR | SAP | Y | Tandem |
| Gomez[71] (P) | Medtronic 640 G and Enlite sensor | Colombia | NRP | 6–79 | – | 55 | 3 | HbA1c | NA | Y | |
| Abraham[64] PLGM (P) | Medtronic 640 G and Enlite sensor | Australia | RCT | 8–20 | <10 | 169 | 6 | Hypoglycemia | SAP | Y | JDRF.au, Australian Research Council Special Research Initiative, in-kind support Medtronic |
| Bosi[67] SMILE (P) | Medtronic 640 G and Enlite sensor | UK, Canada, France, Italy, Netherlands | RCT | 24–75 | 5.8–10 | 153 | 6 | Hypoglycemia | SMBG/pump | Y | Medtronic |
| *Decision support tools* | | | | | | | | | | | |
| Nimri[47] ADVICE4U (P) | AI-DSS; The DreaMed Advisor Pro | USA, Germany, Israel, Slovenia | RCT | 10–21 | 7–10 | 138 | 6 | TIR | Physician guidance | Y | HCT |
| Bisio[46] (P) | Dexcom G5 and inControl Advice (TypeZero Technologies) | USA | RCT (2:1) | ≥15 | – | 80 | 3 | TIR | CGM/MDI | N | Abbott |

[a]All trials, if randomized, were randomized 1:1 unless the randomization scheme is listed in this column. *RCT* randomized controlled trial, *Rcrossover* randomized crossover trial, *NRP* non-randomized prospective.
[b]Ages 4–14 required an enrollment HbA1c ≤ 8.5% and ages 15–45 HbA1c ≤ 8.0%.
*SMBG* self-monitoring blood glucose, *MDI* multiple daily injection.
*PRO* patient-reported outcome, *NH* nocturnal hypoglycemia.
*TIR* time in range (70–180 mg/dL), *TBR* time below range (<70 mg/dL).
*UACR* urine albumin:creatinine ratio.

concealment, and blinding of outcome assessment were not reported by either of the articles. Both study personnel and participants were aware of the treatment groups. Both articles had complete data and minimal bias for selective reporting. (GRADE Evidence: Level B).

**Insulin pump and sensor-augmented pump therapy**. First tested in the late 1970s, CSII with or without CGM provides a more physiologic means of insulin delivery[48–50]. Registry data has demonstrated increased use of pumps in clinical practice over time[45,51–53]. We identified nine relevant articles meeting our inclusion criteria (Table 2). Only two articles reported primary analysis results from an RCT[54,55]. A secondary analysis of an RCT that focused on the pediatric cohort was also identified[56]. Another study was an extension analysis that assessed metabolic outcomes one year after the end of the European multicenter randomized Pediatric Onset Study[57]. The remaining five articles described PROs from RCTs[58–62]. The majority of these trials (77%) were industry-sponsored.

The sample sizes of these studies ranged from 60 to 485 individuals. The study duration ranged from 6.5 to 24 months, and the primary outcomes were TIR[54], HbA1c[57,59], urine albumin-to-creatinine ratio (UACR)[55], and participant satisfaction[58,60,61]. Beck et al. reported that in adults with type 1 diabetes using CGM, initiation of insulin pump therapy improved TIR at the expense of increased hypoglycemia and without significant change in HbA1c[54]. Kordonouri et al. showed that in participants with sensor-augmented pump (SAP) therapy with frequent CGM sensor use at 24 months, there was significantly less C-peptide loss, though changes in HbA1c were not significant[57]. Tanenberg et al. demonstrated a greater decline in HbA1c with more frequent CGM sensor use in both adult and pediatric participants with type 1 diabetes on SAP[59]. The next report compared the effect of SAP to MDI on the UACR in individuals with previous or current albuminuria[55]. In this study, the primary outcome was not met, SAP did not significantly decrease UACR[55]. Speight et al. reported increased satisfaction regarding the delivery device and hypoglycemic control with insulin pump compared with MDI at 6 months, but this difference did not persist at 2 years[58].

Information on sequence generation (3/9), allocation concealment (2/9), and blinding of outcome assessment (1/9) was only reported in a few studies. In the nine trials, both study personnel and participants were aware of the treatment groups. Only one study did not have information on selective reporting, and 4/9 reported data completeness; all other studies had minimal risk for bias. (GRADE Evidence: Level B)

**Low glucose suspend and predictive low glucose suspend**. The ability to pair CGM with insulin pumps to control insulin infusion rates led initially to two approaches to minimize the risk for hypoglycemia[63]. Low glucose suspend (LGS) algorithms shut off insulin delivery when continuously monitored glucose levels reach a pre-set hypoglycemic threshold. Predictive LGS (PLGS) algorithms analyze dynamic trends in glucose levels and shut off insulin delivery when hypoglycemia is imminent but before hypoglycemic thresholds are achieved. Both approaches allow insulin delivery to resume when glucose levels return to pre-specified thresholds. These two approaches, implemented using a variety of CGM and pump platforms, have been tested in many robust RCTs over the last decade.

Among the 11 studies identified[64–74], 10 were RCTs, including seven parallel arm comparisons and three with a cross-over design (Table 2). Five of the trials were funded by device manufacturers, with an additional 3 trials receiving in-kind support. One non-randomized observational study was identified[71]. The period of observation varied in the studies from 2 weeks to as long as 12 months. Altogether, 1130 participants were included in the trials, which ranged in size from 55 to 247 participants. Eight reported primary results[64–68,70–72] and 3 were secondary analyses[69,73,74]. In 7 of the 8 primary reports, hypoglycemia events or TBR were the primary outcome[64,65,67,68,70,72]. All of these trials demonstrated significant improvements in measures of hypoglycemia with either a PLGS or LGS system[64,65,67,68,70,72]. In one trial, change in HbA1c was the primary outcome, and the PLGS arm was found to be superior to the control group[71]. Collectively, the findings of these trials indicate that LGS/PLGS results in statistical and clinically significant improvements in the risk of hypoglycemia.

Limitations of data include the fact that several different algorithms for suspending insulin were tested, reflecting the rapid technological evolution of the field. As with many other trials in the field, these studies are also limited by the lack of diversity of the populations studied, including diversity in race/ethnicity, access to care, and severity of concomitant diabetes complications and other comorbidities. Of the articles included, 5/11 had a low risk of bias for sequence generation, and 3 had a low risk of bias for allocation concealment. Blinding of outcome assessment could only be assessed in a single study. Most of the trials had complete data and low bias for selective reporting. (GRADE evidence: Level B).

**Automated insulin delivery**. Building on the strategy of using sensor glucose data to suspend insulin delivery for low or falling glucose levels, developers of automated insulin delivery (AID) systems, also known as closed-loop control or artificial pancreas, included programming to increase insulin delivery to mitigate hyperglycemia[75]. AID systems consist of a glucose sensor, which drives an insulin pump to alter insulin delivery based on algorithmically determined modulations. Early feasibility studies explored the use of a fully autonomous system; however, rises in postprandial glucose with this approach led to the use of a hybrid approach whereby the user 'announces' meals prior to eating. There has been rapid iteration of systems and innovation leading to the testing of multiple pump and sensor combinations with various algorithms forming the backbone of AID.

As demonstrated in Table 3, 21 articles on AID were identified, with Fig. 4 highlighting the increase in investigations of this technology over time. Eighteen articles reported primary findings from an RCT and encompassed 1752 participants with individual studies ranging from 52 to 172 participants[76–93], two articles providing information on secondary outcomes[94,95], and one from an extension study[96]. The vast majority of studies had a duration of 3–6 months, though some early feasibility studies lasted only a few days[89,93] while another study that assessed the impact of using AID shortly after diagnosis, followed by SAP on preserving beta-cell function used a year-long intervention[82]. Amongst the studies reported, one-third were industry-sponsored, and in-kind support was noted in 5 of the remaining trials.

Based upon the shift that occurred following the standardization of metrics to be used for CGM data as well as consensus targets that have been developed[11,97], the primary outcome was TIR in over half the primary trials. The mean adjusted difference in TIR between study groups ranged from 9 to 15%, which equates to 2¼–3½ more hours a day of TIR[76,77,80,81,83,86,91,92].

Seminal RCT studies of the Tandem t:slim X2 with Control IQ demonstrated increased TIR in both adolescents and adults[81] and children aged 6–13 years[80]. Recognizing that adolescents and young adults often have the greatest difficulty achieving glycemic targets, Bergenstal and colleagues randomized 113 participants aged 14–29 years to a commercially available AID system (Medtronic 670 G) followed by an investigational device

**Table 3 AID-related clinical trial publications included in the systematic review (2012-2022).**

| Author; year of publication; trial cohort if applicable; analysis type: primary (P), secondary (s), extension (E), (PRO) | System or product studied | Locations (trial sites) | Study design[a] | Age range (yrs) | HbA1c (%) range | # Random-ized | Trial duration (mos) | Primary outcome | Comparator/ assessed measure | Sig (Y/N) | Funding |
|---|---|---|---|---|---|---|---|---|---|---|---|
| Buckingham[82] DirecNet/ TrialNet (P) | Hybrid closed-loop control: Medtronic Paradigm or Revel with Sof-sensor | USA | RCT (2:1) | 6-45 | NA | 68 | 12 | C-peptide | Usual care | N | NIH-NICHD |
| Phillip[89] MD-Logic Camp (P) | MD-Logic Artificial Pancreas system: Medtronic Paradigm Veo and Enlite sensor | Germany, Israel, Slovenia | Rcrossover | 10-18 | 7-10 | 56 | 0.067 | Hypoglycemia | SAP | Y | Sanofi, Slovenian National Research Agency, in-kind support Medtronic, Intel Israel, Dell Israel, Yael Software and Systems |
| Russell[93] Bionic Pancreas (P) | Bionic pancreas: Tandem tslim X2 and Dexcom G4 | USA | Rcrossover | ≥12 | – | 52 | 0.33 | Plasma glucose | Usual care | Y | NIH-NIDDK, HCT, Earle Charlton Fund for Innovative Research in Diabetes, Frederick Banting Foundation, Ralph Faber, and John Whitlock |
| Thabit[92] APCam8; AP@home04 (P) | FlorenceD2A closed-loop system: Dana R Diabecare pump and Abbott FreeStyle Navigator II | Austria, Germany, UK | Rcrossover | ≥6 | 7.5-10 or <10[b] | 58 | 3 | TIR | SAP | Y | Eli Lilly and Sanofi, in-kind support Medtronic |
| Nimri[88] MD-Logic Home (P) | MD-Logic Artificial Pancreas system: Medtronic Paradigm Veo and Enlite sensor | German, Israel, Slovenia | Rcrossover | 10-65 | 7-9 | 75 | 1.5 | TBR | SAP | Y | Sanofi, in-kind support Medtronic |

**Table 3 (continued)**

| Author; year of publication; trial cohort if applicable; analysis type: primary (P), secondary (s), extension (E), (PRO) | System or product studied | Locations (trial sites) | Study design[a] | Age range (yrs) | HbA1c (%) range | # Random-ized | Trial duration (mos) | Primary outcome | Comparator/ assessed measure | Sig (Y/N) | Funding |
|---|---|---|---|---|---|---|---|---|---|---|---|
| Tauschmann[91] APCam11 (P) | FlorenceM closed-loop system: Medtronic modified 640 G, Enlite 3 sensor, and Cambridge algorithm | USA, UK | RCT | ≥6 | 7.5–10 | 86 | 3 | TIR | SAP | Y | JDRF, NIHR Cambridge Biomedical Research Center, Wellcome Strategic Award |
| Brown[81] iDCL (P) | Tandem tslim X2 with Control IQ and Dexcom G6 | USA | RCT (2:1) | ≥14 | Any | 168 | 6 | TIR | SAP | Y | NIH-NIDDK |
| Benhamou[77] Diabeloop WP7 (P) | Diabeloop Generation 1: Cellnovo patch-pump with Diabeloop application and Dexcom G5 | France | Rcrossover | ≥18 | ≤10 | 68 | 3 | TIR | SAP | Y | French Innovation Fund, Diabeloop |
| Kovatchev[85] Feb Project Nightlight (P) | Tandem tslim X2 with Control IQ and Dexcom G6 | USA | Rcrossover | 18–70 | – | 93 | 2.5 | TBR | SAP | Y | NIH-NIDDK |
| Breton[80] iDCL (P) | Tandem tslim X2 with Control IQ and Dexcom G6 | USA | RCT | 6–13 | 5.7–10 | 101 | 4 | TIR | SAP | Y | Tandem, NIH-NIDDK |
| Kovatchev[84] iDCL (P) | Tandem tslim X2 with Control IQ and Dexcom G6 | USA | RCT | ≥14 | <10.5 | 127 | 3 | TBR | SAP | Y | NIH-NIDDK |
| McAuley[86] JDRF.au Closed-Loop (P) | Medtronic 670 G and Enlite 3 sensor | Australia | RCT | 25–70 | ≤10.5 | 120 | 6 | TIR | Pump/MDI | Y | JDRF.au, National Health and Medical Research Council (NHMRC) of Australia |
| Bergenstal[78] FLAIR (P) | DreaMed MD-Logic artificial pancreas algorithm with Medtronic 670 G and Guardian 3 sensor | USA, Germany, Israel, Slovenia | Rcrossover | 14–30 | 7–11 | 113 | 6 | TAR | AID algorithm | Y | NIH-NIDDK |
| Kanapka[96] iDCL (E) | Tandem tslim X2 with Control IQ and Dexcom G6 | USA | RCT | 6–13 | 5.7–10 | 101 | 7 | TIR | SAP | Y | Tandem, NIH-NIDDK, in-kind |

**Table 3 (continued)**

| Author; year of publication; trial cohort if applicable; analysis type: primary (P), secondary (s), extension (E), (PRO) | System or product studied | Locations (trial sites) | Study design[a] | Age range (yrs) | HbA1c (%) range | # Randomized | Trial duration (mos) | Primary outcome | Comparator/assessed measure | Sig (Y/N) | Funding |
|---|---|---|---|---|---|---|---|---|---|---|---|
| Collyns[83] (P) | Medtronic 670 G and Guardian 3 sensor | New Zealand | Rcrossover | 7-80 | <10 | 60 | 3 | TIR | PLGS | Y | support Tandem Medtronic, Otago University |
| Abraham[76] JDRF.au Closed-Loop (P) | Medtronic 670 G and Guardian 3 sensor | Australia | RCT | 12-25 | ≤10.5 | 172 | 6 | TIR | Usual care | Y | JDRF.au, Special Research Initiative of NHMRC, PCH Foundation, in-kind support Medtronic, Roche Diabetes Care |
| Choudhary[79] ADAPT (P) | Medtronic 780 G and Guardian 3 sensor | UK, France, Germany | RCT | ≥18 | ≥8 | 82 | 6 | HbA1c | CGM/MDI | Y | Medtronic |
| Renard[87] FREELIFE-KID (P/E) | Tandem tslim X2 with Control IQ and Dexcom G6 | France | RCT | 6-12 | <10 | 120 | 4.5 | TIR | 24/7 vs E/N | Y | French Health Ministry |
| Ware[90] DAN05 (P) | (1) a modified Medtronic 640 G and Guardian 3 sensor (FlorenceM), (2) Dana Diabecare RS pump and Dexcom G6 (CamAPS FX) | USA, UK | RCT | 6-18 | 7-10 | 133 | 6 | HbA1c | Pump | Y | NIH-NIDDK, NIHR Cambridge Biomedical Research Center and Wellcome Strategic Award, in-kind support Abbott, Dexcom, Medtronic |
| Ekhlaspour[94] April iDCL (S) | Tandem tslim X2 with Control IQ and Dexcom G6 | USA | RCT (2:1) | ≥14 | Any | 168 | 6 | TIR | Usual care | N | NIH-NIDDK, in-kind support Tandem |
| Ekhlaspour[95] Aug iDCL (S) | Tandem tslim X2 with Control IQ and Dexcom G6 | USA | RCT (2:1) | ≥14 | Any | 168 | 6 | TIR | By HbA1c | Y | NIH-NIDDK, in-kind support Tandem |

aAll trials, if randomized, were randomized 1:1 unless the randomization scheme is listed in this column. *RCT* randomized controlled trial, *RcrossoverT* randomized crossover trial, *NRP* non-randomized prospective.
bAges ≥18 required an enrollment HbA1c 7.5-10% and ages 6-18 HbA1c < 10%.
*MDI* multiple daily injection, *TAR* time above range (>180 mg/dL).
*TIR* time in range (70-180 mg/dL), *TBR* time below range (<70 mg/dL).

(advanced hybrid closed loop similar to Medtronic 780 G), or vice versa[78]. Both times above the target range (>180 mg/dL; TAR) during the day and time <54 mg/dL over 24 h reached statistical significance[78]. In studies where HbA1c was the primary outcome, whether assessing the use of the Medtronic 780 G in adults who were previously managed with MDI and isCGM[79] or in 6–18-year-olds using the CamAPS FX AID system,(84) AID systems were found to be superior to the comparator groups.

For quality assessment, sequence generation had a low risk of bias in primary manuscripts (not reported in 9/18). The majority of studies did not provide details on either allocation concealment (11/18) or blinding of outcome assessment (13/18). Participants and study personnel were aware of the randomized intervention as it is not feasible to be blind to the use of the AID system. Complete data were noted in the trials with minimal bias for selective reporting. (GRADE Evidence: Level B).

Despite the rigorous procedures used, some AID trials that met inclusion criteria were not identified. For example, Ware et al. described the use of the CamAPS FX application in children aged 1-7 years old[98]. In 74 participants, mean age of 5.6 ± 1.6 years, undergoing 16 weeks of AID, then SAP or vice-versa, TIR rose 8.7% with AID use without increasing TBR, demonstrating the beneficial impact that this AID system can have on very young children[98]. Additionally, since the last literature search was conducted on 9/5/2022, additional trials have been completed. The pivotal trial describing the use of the insulin-only bionic pancreas[99], as well as secondary analyses of this cohort, which examined system use in pediatric and adult cohorts[100,101], and results from the extension phase of the trial were published[102]. Reports on open-source AID systems have documented the efficacy of these do-it-yourself (DIY) systems both in the primary trial[103] and with continued use during an extension phase[104]. Two studies have investigated whether attaining strict glucose control through the use of an AID system could preserve residual beta-cell function as measured by C-peptide secretion, both failing to show a difference compared with those receiving standard of care[105,106].

Continued exploration regarding the use of AID systems in subgroups of individuals living with diabetes, like very young children[107], echo the findings regarding the beneficial impact of AID use. For systems initially approved with single-arm studies, post-marketing RCTs have demonstrated the benefits of system use[108].

**Person-reported outcomes**. PROs capture the subjective experience of a person and are obtained through vetted and validated methods such as questionnaires and interviews. PROs offer a layer of understanding of the impact of devices and technologies that go beyond common glycemic outcomes such as HbA1c and TIR. They are important because even devices that offer glycemic or health benefits can be discontinued by the user if the devices are additionally burdensome or distressing to the person. Further, knowledge of a device's specific impact on the person's quality of life offers an additional clinical variable to consider when deciding on the type of device to recommend or prescribe. In our review of included PROs, we identified nine studies that had validated PROs measurements[36–38,58–62]. In all the cases, PROs served as a secondary (or beyond) endpoint, with glycemic outcomes (commonly TIR) as the primary endpoint. Of these nine studies, six were RCTs[38,58–61], and two were prospective cohort studies (no control group)[27,62]. Other studies on PROs were excluded because they were not part of testing of a specific device's clinical efficacy or effectiveness. The devices tested in these nine studies included CGM, LGS, and SAP. These studies cut across pediatric and adult samples, with most focused on

those with established diabetes (i.e., few newly diagnosed individuals included).

PROs measured in the studies varied and included self-efficacy about device use, satisfaction with and benefits from the device, diabetes distress, and quality of life. They utilized validated surveys with diabetes distress, one of the most common areas examined. This is consistent with PROs evaluated in studies of people with diabetes—diabetes distress is a common and uncomfortable experience for people with diabetes and an area that negatively impacts self-management and outcomes. In five of the studies, at least one PRO was statistically different from the comparison (control group or non-users of the device) in a favorable direction.

The quality of the studies was relatively high given the randomized and controlled nature in most cases and validated, commonly used PROs measurements. Participants were aware of the PROs measurements, and they were not asked about specific brands of devices as they completed questionnaires. A number of the studies were funded by the device manufacturer, but the nature of an external ethical review and data management coordinating center kept potential bias low. Overall, evidence from these studies considering the high quality indicates that devices can positively impact PROs. (GRADE Evidence: Level B).

It is not presently known which PRO is most likely to show benefit and if it will happen across all devices and across various subgroups of those living with type 1 diabetes, for example, related to age, duration of diabetes, or previous diabetes management approach. Thus, the inclusion of PROs in studies on devices will continue to extend our understanding of the amount, type, and timing of impact to help direct a precision medicine approach to care.

**Precision assessment**. Living with and managing type 1 diabetes is an exquisitely individualized experience that is influenced by many internal and external factors such as age, therapy goals, caregiver needs, and many more. Thus, it is not surprising that the use of technology in the management of type 1 diabetes must also be individualized. The purpose of this systematic review was to assess diabetes technologies using a precision medicine lens. In the previous sections, we critically reviewed the efficacy of technologies and the quality of these clinical trials; herein, we will focus on any precision-based analyses or specific populations studied within these trials and the quality of such analyses.

Of the 70 articles included in our review, 28 (40%) included some form of precision-focused analysis (Tables 4–6)—either post-hoc comparisons of baseline features within or between treatment arms or a trial with a narrowly defined enrollment population (e.g., ages 4–9 years). Baseline characteristics evaluated included age at enrollment, diabetes duration, sex, race/ethnicity, insulin modality (MDI versus CSII), metabolic status (HbA1c, C-peptide, hypoglycemia unawareness status), BMI, presence of nephropathy, education level, and household income. Additional post-treatment variations in efficacy were noted, such as differences in the degree of technology engagement or daytime versus nighttime glycemic outcomes. Characterization of these results could benefit people with diabetes and their care teams when seeking to maximize benefits and individualize the approach to technology.

We found the majority of analyses that included pediatric and adult cohorts revealed similar results in metabolic outcomes, though there were some notable exceptions. It is important to remember that these are not direct comparisons of age groups but are post-hoc subgroup analyses of the primary and secondary trial endpoints among those of a specific age range. For example, adults enrolled in these clinical trials, even stratified within ages 18 to 87 years, had significant metabolic benefits from CGM[14,17,22,35],

**Table 4 CGM articles with precision-directed analyses or subpopulations of interest.**

| Author year (trial/cohort title) location | Analysis type/study design/number of randomized | Primary endpoint (SIG vs. NS)—% dropout multiple test correction performed? | Subgroup analyses—technology superior to standard of care | Subgroup analyses—technology comparable to standard of care |
|---|---|---|---|---|
| Battelino[22] (SWITCH) Europe | Primary Randomized crossover 153 | HbA1c 6 months (SIG)—0% Unsure | Age Mean HbA1c difference in ages 6-18 years (n = 72) −0.26% to −0.66% (95% CI: −0.26% to −0.66%; p < 0.001) and in ages 19-70 years (n = 81) −0.41% (95% CI: −0.28% to −0.53%; p < 0.001). | Age |
| Mauras[25] (DirecNet) USA | Primary RCT 137 | HbA1c reduction ≥0.5% a6 months (NS)—6% Unsure | Sensor wear Of 28 children who wore the sensor ≥ 6 days/week during month 6, reduction in HbA1c compared with the 41 who did not (HbA1c change −0.3 ± 0.7% vs. 0.0 ± 0.5%, p = 0.01). | Age, sex, race/ethnicity, parent education level, MDI vs. pump, HbA1c, BMI No difference in HbA1c between CGM and SMBG (ages 4-9 years: 33 age 4-5; 43 age 6-7; 61 age 8-9). |
| Laffel[17] (CITY) USA | Primary RCT 153 | Δ HbA1c 6 months (SIG)—7% Yes[a] | Age <19 vs. ≥19 years, HbA1c < 9 vs. ≥9%, MDI vs. pump, sex, race, detectable C-peptide, prior CGM use Entire cohort (ages 14-24 years) had an HbA1c difference of −0.37% (95% CI: −0.66%, to −0.08%; p = 0.01). | |
| VanName[32] (SENCE) USA | Extension RCT 143 | TIR 6 months (NS)—4% Yes[a] | Age In ages 2-7 years reduction in TBR during extension study within groups (CGM+family behavioral intervention: −1.4%, CGM: −2%, SMBG-to-CGM: −2.8%; p < 0.001 for all). | Age In ages 2-7 years, there was no significant increase in TIR (about +3%) or decrease in HbA1c (about −0.1%) during extension study within or between groups. |
| Raviteja[15] India | Primary RCT 68 | Δ HbA1c 3 months (NS)—7% Unsure | HbA1c Children with HbA1c > 7.5% at baseline demonstrated a significant decrease in HbA1c (−1.27% ± 1.46 vs. 0.13% ± 1.76; p = 0.045). | Age, HbA1c |
| Miller[35] (WISDM) USA | Extension RCT 203 | Δ TBR 6 months (SIG)—2% Yes[a] | Age, MDI vs. pump, daytime vs. nighttime Seniors (aged 60-87 years) on CGM had reduced TBR (−3.4%, −3.1%) and HbA1c (+8%, +4%) and increased TIR (−0.2%, −0.2%); p ≤ 0.01 for all. | |
| vanBeers[14] (IN CONTROL) Netherlands | Primary Randomized crossover 52 | TIR 8 months (SIG)—12% Unsure | IAH, MDI vs. pump, carbohydrate counting vs. not Adults (aged 18-75 years) with IAH had improved TIR (+9.6% [95% CI: 8.0-11.2%]; p < 0.0001) and fewer SH events (14 vs. 34, p = 0.033) | In ages 2-10 years there was no significant decrease in HbA1c using CGM or with baseline HbA1c < 7.5%. |
| Beck[21] (DIaMonD) USA | Primary RCT 158 | Δ HbA1c 6 months (SIG)—2% Yes | Age, HbA1c, TBR, SMBG frequency, education level, IAH, MDI only, diabetes numeracy, and hypoglycemia fear scores HbA1c improvement in adults, ages 26-73 years on MDI (−0.6% [95% CI: −0.8% to −0.3%]; p < 0.001). | |
| Heinemann[18] (HypoDE) Germany | Primary RCT 149 | # hypoglycemic events 6 months (SIG)—5% Unsure | IAH, MDI only Hypoglycemia events decreased (IRR 0.28 [95% CI: 0.20-0.39]; p < 0.0001) in ages ≥18 years with IAH on MDI. | |
| Oskarsson[29] (IMPACT) Europe | Secondary RCT 163 | Δ TBR at 6 months (SIG)—1% No | MDI only, HbA1c TBR improved in adults, ages ≥18 years, on MDI with baseline HbA1c ≤ 7.5% (−1.65% [95% CI: -2.21 to -1.09%]; p < 0.0001). | |

[a]FDR adjustment using Benjamini Hochberg procedure.

**Table 5 LGS/PLGS articles with precision-directed analyses or subpopulations of interest.**

| Author year (trial/ cohort title) location | Analysis type/study design/ number randomized | Primary endpoint (SIG vs. NS)—% dropout multiple test correction performed? | Subgroup analyses—technology superior to standard of care | Subgroup analyses—technology comparable to standard of care |
|---|---|---|---|---|
| Beck[54] (DIaMonD) USA | Primary RCT 75 | Δ TIR 7 months (SIG)—5% Unsure | *HbA1c* Age < 50 vs. ≥50, TIR < 53 vs. ≥53%, TBR < 3 vs. ≥ 3%, HbA1c, SMBG frequency, education level, IAH, Hypoglycemia Fear Adults on CGM + pump (vs. CGM+MDI), aged 26–73 years, had improved TIR (+83 min [95% CI: 17–149]; p = 0.01). | *HbA1c* Baseline HbA1c did interact (p = 0.006) with the intervention on the outcome where HbA1c <7.5% did not improve TIR. |
| Bergenstal[66] (ASPIRE) USA | Primary RCT 247 | NH events AUC (SIG)—3% Unsure | Age 16–24 vs. 25–50 vs. 51–70, HbA1c ≤ 7 vs. >7%, diabetes duration ≤ 25 vs. >25 years In the cohort (ages 16–70 years), LGS reduced NH & HbA1c compared to SAP. | |
| Rosenlund[55] Denmark | Primary RCT 60 | Δ UACR 12 months (NS)—8% Unsure | *Complication* Complication, sex, BMI, HbA1c SAP (vs. MDI) improved UACR in those with current or past albuminuria when adjusting for HbA1c, sex, and BMI (p = 0.02). UACR improved in those with current albuminuria (n = 48; −18 vs. +38%, p = 0.011). | *Complication* SAP-treated individuals with current or past albuminuria (aged 18–75 years) did not have statistical improvement in UACR over MDI (−13 vs. +30%, p = 0.051). |
| Weiss[73] (APSIRE) USA | Secondary RCT 247 | NH events AUC (SIG)—3% Unsure | *HbA1c* Reduced NH events for participants with a baseline HbA1c < 7% (n = 95) and 7–8% (n = 115). NH event glucose nadir (mg/dL) was higher if HbA1c < 7% (49.7 ± 8.5 vs. 46.8 ± 9.6, p < 0.001) or 7–8% (49.9 ± 8.2 vs. 48.1 ± 9.6, p = 0.007). | *HbA1c* NH events not reduced in small group with HbA1c > 8% (n = 30). NH glucose nadir (mg/dL) was similar between LGS and SAP if HbA1c > 8% (49.5 ± 8.6 vs. 46.5 ± 10.1, p = 0.06). |
| Slover[56] (STAR 3) USA | Secondary Randomized crossover 156 | HbA1c at 1 year (SIG)—0% Unsure | *Age* In the entire cohort (ages 7–18 years), HbA1c was reduced in SAP vs. MDI. Those aged 7–12 years (n = 82) had increased sensor wear compared to ages 13–18 (n = 74; p = 0.025). | *Age* Secondary metabolic endpoints were similar between age groups, except MAGE did not improve, and BMI rose in ages 13–18 years. |
| Ly[72] Australia | Primary RCT 95 | Mod/sev hypoglycemia 6 months (SIG)—9% Unsure | *Age* LGS improved hypoglycemia in the entire cohort (ages 4–50 years; IRR 3.6 [95% CI: 1.7–7.5]; p < 0.001) and in those ages 4–11 years (n = 30; IRR 5.5 [95% CI: 2.0–15.7]; p < 0.001). | |
| Buckingham[68] (IN HOME CLOSED LOOP) North America | Primary Randomized crossover 81 | Median TBR over 42 nights (SIG)—5% No | *Age* PLGS reduced TBR in the entire cohort (ages 4–14 years), specifically in ages 4–10 years by 50% (6.2–3.1%) and in ages 11–14 years by 54% (10.1–4.6%; p < 0.001 for both). | |
| Calhoun[69] (IN HOME CLOSED LOOP) North America | Secondary Randomized crossover 127 | Nights with hypoglycemia (SIG)—0% Yes | *Age* Age, sex, HbA1c, diabetes duration, daily % basal, TDD, bedtime BG, bedtime snack, IOB/TDD, weekday vs. weekend, exercise intensity, number of CGM values ≤ 60 mg/dL from 12–8PM, CGM rate of change before activation, time system activated Night use of PLGS reduced hypoglycemia in the entire cohort (ages 4–45 years) (OR 1.91 [99% CI: 1.57–2.32]; p < 0.001). | *Age* Secondary metabolic endpoints were similar between age groups except TAR, AM glucose, and HbA1c, which were not improved in ages 4–10 years (n = 36). |
| Abraham[64] (PLGM) Australia | Primary RCT 154 | % Time <63 mg/dL at 6 months (SIG)—9% Unsure | *Age* Age, sex, HbA1c, IAH In the entire cohort (ages 8–20 years), percent time with hypoglycemia (<63 and <54 mg/dL) in PLGS vs. SAP was 2.6% vs. 1.5% (p < 0.0001 for both cutoffs). | |

**Table 6 AID Articles with Precision-Directed Analyses or Subpopulations of Interest.**

| Author year (trial/cohort title) location | Analysis type/study design/number of randomized | Primary endpoint (SIG vs. NS)—% dropout multiple test correction performed? | Subgroup analyses—technology superior to standard of care | Subgroup analyses—technology comparable to standard of care |
|---|---|---|---|---|
| Thabit[92] AP@home04 Europe | Primary Randomized crossover 58 | TIR/time in a tight range overnight at 3 months (SIG)—0% No | Age. AID use in adults (aged ≥18 years) and AID use at night in children (aged 6-17 years) improved time in the target range ($p < 0.001$ for both). | Age. Secondary nighttime endpoints were similar between age groups except for TBR, time <50 mg/dL, and number of nights with glucose <63 mg/dL, which were not improved in ages 6-17 years ($n = 25$). |
| Tauschmann[91] (APCam11) USA, UK | Primary RCT 86 | TIR 3 months (SIG)—0% No | Age <13 vs. 13-21 vs. ≥21, sex, HbA1c < 8.5 vs. ≥8.5%. AID use in the entire cohort (aged ≥6 years) increased TIR by 10.8% (95% CI: 8.2-13.5%; $p < 0.0001$) | |
| Collyns[83] New Zealand | Primary Randomized crossover 60 | TIR 3 months (SIG)—2% No | Age. AID use increased TIR in ages 7-13 years ($n = 19$; +11.8 ± 7.4%), ages 14-21 ($n = 14$; +14.4 ± 8.4%), and ages 22-80 ($n = 26$; +11.9 ± 9.5%; $p < 0.001$ for all ages). | |
| Ware[90] (DAN05) USA, UK | Primary RCT 133 | Δ HbA1c 6 months (SIG)—8% Yes[a] | Age, device, device use. AID use in children, aged 6-18 years, improved HbA1c (−0.32% [95% CI: −0.59 to −0.04]; $p = 0.023$). CamAPS FX ($n = 46$) showed an HbA1c change of −1.05% [95% CI: −1.43 to −0.67; $p < 0.0001$] with median 93% device use. | Device, device use. FlorenceM ($n = 75$) showed an HbA1c change of +0.21% (95% CI: −0.14 to 0.57; $p = 0.23$) with median 40% device use. |
| Benhamou[77] (Diabeloop WP7) France | Primary Randomized crossover 68 | TIR 3 months (SIG)—7% No | HbA1c. Increased TIR for ages ≥18 years (+9.2% [95% CI: 6.4-11.9]; $p < 0.0001$) and across HbA1c groups: <7% ($n = 16$), 7-7.4 ($n = 10$), 7.5-7.9 ($n = 18$), 8-8.4 ($n = 8$), and ≥8.5 ($n = 11$; $p < 0.05$ for all) | |
| Breton[80] (iDCL) USA | Primary RCT 101 | TIR 4 months (SIG)—1% Unsure | Age, sex, BMI, household income, parental education, HbA1c. AID increased TIR in ages 6-13 years by +11% (95% CI: 7-14; $p < 0.001$). | |
| Kovatchev[84] (iDCL) USA | Primary RCT 127 | TBR & TAR 3 months (SIG)—2% Yes | Age, sex, BMI, HbA1c, C-peptide, TBR, TAR, previous CGM use, household income, education. AID reduced TBR (−1.7% [95% CI: −2.4 to −1.0]; $p < 0.0001$ superiority) & TAR (−3.0% [95% CI: −6.2 to 0.1]; $p < 0.0001$ noninferiority) in ages ≥14 years. | Sex, TBR. In male participants, there was no change in TAR ($p_{interaction} = 0.018$) compared to −11% in females ($n = 59$). Those with baseline TBR ≤4.0% ($n = 63$) had less improvement in TBR (−1%) ($p_{interaction} = 0.0406$) TBR > 4.0% (−3%). |
| Ekhlaspour[94] (iDCL) USA | Secondary RCT 168 | TIR 6 months (SIG)—0% No | Age, device use. AID increased TIR in ages 6-13 years by +2.6 hours with no significant interaction with baseline device use for TIR or TBR. | |
| McAuley[86] (Australian JDRF Closed-Loop) | Primary RCT 120 | TIR 6 months (SIG)—8% Unsure | Sex, insulin modality prior to trial. AID increased TIR in ages 25-70 years by +15% (95% CI: 11-19; $p < 0.0001$) similarly for males/females, MDI/pump users. | |

aFDR Adjustment using Benjamini Hochberg procedure, IAH impaired awareness of hypoglycemia.
SH severe hypoglycemia, IRR incidence rate ratio.
NH nocturnal hypoglycemia, RAS renin-angiotensin system.
MAGE mean amplitude of glucose excursion (mean of blood glucose values exceeding 1 SD from the 24-h mean blood glucose, used as an index of glycemic variability).

SAP[18,54], LGS/PLGS[64,72], and AID systems[77,83–86,91,92]. Older adults with type 1 diabetes (duration 1–71 years) were studied by Miller et al. and had improvements in TBR, TIR, and HbA1c even in the extension phase of the RCT[35]. Three RCTs in adults using MDI only demonstrated improvement in glycemic metrics with CGM use compared to SMBG[14,18,21,109].

Technology RCTs in children, especially very young children, however, have produced mixed results. CGM trials in young children, encompassing ages 2–10 years, failed to meet their primary endpoint of HbA1c reduction. The authors posit this could be due to parental fear of hypoglycemia despite overall parental satisfaction with the technology[15,25,32]. Subgroup analyses showed increased sensor wear[25] or higher baseline HbA1c (>7.5%)[15] in these trials were associated with improvements in HbA1c with CGM over SMBG. Other endpoints, such as TBR[64,68] or hypoglycemia occurrences[69,72], in SAP, LGS, and PLGS trials demonstrated efficacy in children (age range 4–18 years). Adolescents demonstrated worsening of some secondary metabolic endpoints such as increased glycemic variability and increased BMI in an SAP vs. MDI trial[56], no change in hypoglycemia measures in an AID trial where adults demonstrated improvement[92], and males (age 14 years and up) failed to improve TAR while females did[84]. A consistent picture emerged from trials of AID with efficacy (e.g., improved TIR, TBR, TAR, HbA1c) in all ages (ages 6–70 years), even with age stratification and differences in algorithm and platform.

Other demographic features were sparsely reported, especially race and ethnicity. The diversity of most type 1 diabetes trials, including technology trials, is limited. In pediatric trials, parent education level and income were not significantly associated with outcome, but again diversity of these factors may be limited in clinical trials and is an area of need to improve inclusivity[25,80,85]. BMI was only used in stratification or model inclusion in 4 trials[25,55,80,84] and only affected the outcome in the study by Rosenlund et al. in adults with albuminuria[55].

Baseline HbA1c subgroup analyses in most trials produced consistent results except in three studies[15,54,74]. Children with HbA1c > 7.5% (vs. <7.5%) on CGM had improved HbA1c at 3 months[15]. Adults with HbA1c > 7.5% (vs. <7.5%) on SAP had improved TIR[54]. Adolescents and adults with HbA1c < 8% (vs. >8%) on LGS had reduced NH events[74]. The presence or absence of detectable C-peptide did not have any bearing on metabolic outcomes[17,85].

Several studies focused on adults with impaired awareness of hypoglycemia (IAH) and demonstrated improvements in TIR, HbA1c, and rates of hypoglycemia with CGM[14,18,21]. In addition, CGM trials often performed subgroup analyses based on the type of insulin modality used by participants, MDI vs. pump, and this did not affect the efficacy of any results[14,17,35]. Diabetes Numeracy[21] and Hypoglycemia Fear[21,54] assessment via validated questionnaires did not affect the results of these trials. Unsurprisingly, higher CGM sensor and AID wear time were often tied to increased efficacy[25,90,95].

A special population of adults treated for nephropathy with past or current albuminuria was treated with SAP (vs. MDI) for 1 year, and while there was not a significant improvement in UACR, when these results were adjusted for baseline sex, BMI, and HbA1c, this became significant[55]. Additionally, in a subgroup with albuminuria present at screening (UACR ≥ 30 mg/g), there was an 18% reduction in UACR in the SAP group compared to +38% in the MDI group ($p = 0.011$)[55].

These subgroup analyses provide vital, though exploratory, information for people with diabetes and their providers to consider when discussing technology use in type 1 diabetes management. Only 7/28 (25%) included some form of correction for multiple testing, and many were not pre-specified analyses. In this age of precision-directed care, more rigorous statistical methods should be considered for subgroups of interest in all RCTs, especially in regard to technology use. In summary, all ages benefit from technology, and there are not currently head-to-head comparisons of which device may offer the most benefit for an individual. People with diabetes with IAH or on MDI can still derive significant benefits and should be offered technology. Some technologies may lend themselves better to HbA1c improvement and some hypoglycemia reduction, and these can be considered as part of the goals of treatment.

**Discussion**
Our multi-disciplinary international team of experts in the management of type 1 diabetes was charged with identifying and analyzing the current evidence regarding a precision medicine approach to the treatment of type 1 diabetes. Given the wide-ranging breadth of treatment approaches to type 1 diabetes, we elected to focus on the modern topic of advanced diabetes technologies and limited our literature review to the past decade. We identified 70 peer-reviewed publications that fulfilled the pre-specified criteria of including people with type 1 diabetes in RCTs with a minimum of 50 participants, published in English on or after January 1st, 2012, and we included any secondary or extension studies from these RCTs. Six non-randomized prospective studies were included from under-represented countries to increase diversity and generalizability. We still uncovered some newer as well as unexplained omissions of publications germane to our topic, highlighting the dynamic nature of treatment approaches to type 1 diabetes and challenges to systematic review of heterogenous trials and technologies. This collective effort serves to advance the PMDI, with a major goal of identifying remaining gaps and future directions.

Overall, the studies of advanced diabetes technologies yielded promising results with respect to improvements in glycemia measured as HbA1c or CGM TIR without an increase in hypoglycemia and often with reductions in hypoglycemia. Many studies also provided positive results with respect to PROs or, at a minimum, no increase in self-care burden compared with standard care. The evidence from this systematic review was considered Grade B in all areas, given the rigorous nature of the studies with notable limitations related to relatively small sample sizes and short durations of follow-up. The assessment of bias was variable across the different areas of diabetes technologies, mostly due to incomplete information regarding sequence generation, allocation concealment, and blinding of outcome assessment, although bias was considered low or minimal regarding complete data reporting and selective data reporting.

The DCCT can serve as an example of the evolution of treatment in type 1 diabetes and an approach to precision medicine. Upon its release in 1993, intensive insulin therapy was heralded as the new standard of care for all or most people with type 1 diabetes, although the study was limited to participants aged 13–39 years at entry with duration of diabetes of 1–15 years. Further, there were other inclusion criteria that would likely limit generalizability to the broader population living with type 1 diabetes. Yet, intensive insulin therapy remained the standard approach, with only more recent recommendations regarding tailoring glycemic targets to individual needs based upon impaired awareness of hypoglycemia, co-morbidities, or individual characteristics suggestive of reduced survival[11].

The approach regarding diabetes technology use is reminiscent of the earlier literature on intensive insulin therapy from the DCCT. Following these technology studies, clinical practice recommendations have begun supporting the use of diabetes devices, such as CGM, pump therapy, and even AID systems, in

general for people with type 1 diabetes. Many of these technology studies included participants who were more likely to seek advanced diabetes devices, so-called 'uber' users. Furthermore, most studies included predominantly white, non-Hispanic participants with higher education and/or socio-economic status from developed countries, with the studies performed by experienced clinical teams often in major diabetes centers that have resources that may not reflect real-world experience.

Similarly, the frequency of contact with clinical teams dictated by study protocols in the DCCT and many technology trials simply cannot be translated to clinical practice. Reimbursement issues and a limited workforce mean people with diabetes often do not have the same support as they attempt more intensive management, as occurred after the DCCT or integration of technology. Further, it is difficult to parse out the impact of education delivered during these touch points and how this influences success with devices.

While these studies include people with type 1 diabetes from ages 2 to over 70 years of age, and most studies included male and female participants nearly equally, there are numerous limitations with regard to other subgroups of potentially salient characteristics such as varied glycemia at enrollment with few study participants had significantly elevated HbA1c levels, duration of diabetes, BMI, daily insulin requirements, previous use of diabetes devices, education, co-morbidities including diabetes complications like end-stage renal disease or psychiatric disorders including anorexia nervosa, and inclusion of racial/ethnic minorities. Indeed, it has been well-established that disparities exist regarding access to diabetes technologies for those of minority groups; how we overcome these disparities is of critical importance. Thus, there remains a need to evaluate how diabetes technologies should be used within a precision medicine framework regarding subgroups of people with type 1 diabetes, the timing of initiating diabetes devices, requirements for initial training and education, as well as ongoing follow-up and support to sustain technology use. Further, the durability of device use also remains an ongoing area for future study. Not surprisingly, but important to note device manufacturers sponsored nearly half (44%) of the trials conducted, with in-kind support of devices noted in another 18% of the trials identified.

There is a need for the design and evaluation of more real-world experience of the use of advanced diabetes technologies across these many different subgroups and consideration of when the safety of device use needs reconsideration, for example, in an older population where de-prescribing may be warranted or where technology may even have greater benefits. Indeed, examination of particular subgroups, including those with visual impairment or reduced manual dexterity or cognition, may help guide how best to pair glucose monitoring and insulin delivery modality to the needs of an individual. However, such trials are unlikely to be conducted as head-to-head randomized control trials, again highlighting the need for real-world data collection or use of registry data that would include more diverse populations allowing for exploration of a variety of user characteristics. Additionally, real-world evidence may also provide critical information regarding the frequency of device errors/malfunctions noted with routine use over longer time periods. Exploration of clinically relevant outcomes, like frequency of skin irritation/reactions that may stem from exposure to the adhesives on devices, maybe more feasible with a heterogenous group of users outside the focus of pivotal RCTs, which may instead focus on data collection to support commercialization of products.

Data generated from registries may provide some insight into device use among diverse populations. Registries like the Diabetes Prospective Follow-up initiative, a population-based cohort

with individuals from over 500 diabetes centers across Austria, Germany, Luxembourg, and Switzerland, provide data on more than 90% of the population with type 1 diabetes living in those countries. Additionally, leveraging data from these cohorts, it is feasible to explore questions that could never be answered by an RCT. For example, a recent publication noted that the use of CGM in youth can reduce the risk of severe hypoglycemia and diabetic ketoacidosis[110]. Historically, these types of studies would be relegated to GRADE B evidence in the ADA Standards of Care; the question becomes whether this data is more relevant to clinical practice as compared to the highly selected populations enrolled in rigorous RCTs[111,112]. Notably, when focusing on the latest generation of devices available, much of the real-world evidence published echoes the findings of pivotal trials.

Recognizing that treatment success is not merely confined to glycemic metrics, PROs are now gaining traction in clinical trials. While these tools help to better describe potential benefits, or drawbacks, to various technologies, even they have limitations, like the inability to fully assess cognitive burden. Greater psychological benefit may not be identified unless mixed method approaches are utilized, highlighting the need for more research in this area. Qualitative analysis will increase both the breadth and depth of data collected, which may allow greater focus on device burden and diabetes distress experienced by people with diabetes.

Whereas there is evidence of the benefit of the use of technologies in the treatment of people with type 1 diabetes, there also remains a substantial need to ensure health equity in the use of advanced diabetes technologies for glycemic and personal benefits across the population of those with type 1 diabetes to reduce the already recognized disparities in diabetes device use. The use of advanced diabetes technologies remains an ongoing challenge in low and middle-income countries. Despite these many challenges, the current state of diabetes technology use in the treatment of type 1 diabetes has revolutionized care with the ability to attain more targeted glycemia, reduced hypoglycemia, and preserved well-being without additional burdens for people living with type 1 diabetes and their families. There is more work to do to understand and then implement a precision medicine model, but progress is underway.

## Data availability

The data that support the findings of this study are derived from published, peer-reviewed manuscripts. Source data for all figures are available in Supplementary Data 1.

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

## Acknowledgements

We acknowledge several key persons who assisted with the literature search (Krister Aronsson and Maria Björklund from Lund University), instrumental organizational coordination (Chandra Gruberwith the American Diabetes Association and Emily Mixter at the University of Chicago), and methodological consultation (Diana Sherifali and Russell deSouza from McMaster University as well as Deirdre Tobias from Harvard School of Medicine). Training support was provided by NIH NIDDK K08-DK128628 (L.M.J.), K23-DK129821 (M.T.), K23-DK121771 (J.J.W.), and, in Belgium, a senior clinical research fellowship grant Fonds Wetenschappelijk Onderzoek Flanders (P.G.). NIDDK has provided Diabetes Research Center support at each of these institutions: P30-DK045735 (J.L.S.), P30DK116074 (K.K.H. and J.J.W.), P30DK017047 (I.B.H.), P30-DK116073 (P.A.G.), and P30DK036836 (L.M.L.).

## Author contributions

L.M.J. and J.L.S. reviewed and analyzed the data, drafted the paper, finalized the paper, and prepared tables and figures. E.C., A.C., S.M.P., M.H., S.C., M.U., and S.A. collected and reviewed data. M.T., M.J.R., K.K.H., P.A.G., P.G., J.J.W., I.B.H., and R.E.P. reviewed and analyzed the data and drafted the paper. L.M.L. and C.M. reviewed and analyzed the data, drafted the paper, and finalized the paper.

## Competing interests

The authors declare the following competing interests: J.L.S. serves, or has served, on advisory panels for Bigfoot Biomedical, Cecelia Health, Insulet Corporation, Medtronic Diabetes, StartUp Health Diabetes Moonshot, and Vertex. J.L.S. has served as a consultant to Abbott Diabetes, Bigfoot Biomedical, Insulet, Medtronic Diabetes, and Zealand. Yale School of Medicine has received research support for J.L.S. from Abbott Diabetes, JAEB Center for Health Research, JDRF, Insulet, Medtronic, NIH, and Provention Bio. K.K.H. received consulting fees from Cecelia Health. P.A.G. consults or has consulted for Provention Bio and Viacyte. P.A.G. is the co-founder of IM Therapeutics and serves as CMO on the Board as well as being a shareholder. P.A.G. has also received research support from the Helmsley Foundation, Nova Pharmaceuticals, Imcyse, Provention Bio, Intrexon T1D Partners, and Novartis. All payments have been made to the University of Colorado. P.G. serves, or has served, on the advisory panel for Novo Nordisk, Sanofi-Aventis, Boehringer-Ingelheim, Janssen Pharmaceuticals, Roche, Medtronic, Abbott, Ypsomed, and Bayer. P.G. serves, or has served, on the speaker's bureau for Merck Sharp and Dohme, Boehringer-Ingelheim, Bayer, Medtronic, Insulet, Novo Nordisk, Abbott, Roche, VitalAire, and Dexcom. Financial compensation for these activities has been received by KU Leuven. KU Leuven received non-financial support for PG for travel from Sanofi-Aventis, A Menarini Diagnostics, Novo Nordisk, Medtronic, and Roche. All disclosures are unrelated to the present work. IBH has received research funding from Insulet and Dexcom and is a consultant for Abbott, embecta, Lifescan, and Hagar. REP reports consulting fees from Bayer AG, Corcept Therapeutics Incorporated, Dexcom, Hanmi Pharmaceutical Co., Merck, Novo Nordisk, Pfizer, Sanofi, Scohia Pharma Inc., and Sun Pharmaceutical Industries, and grants/research support from Hanmi Pharmaceutical Co., Janssen, Metavention, Novo Nordisk, Poxel SA, and Sanofi. All funds are paid directly to REP's employer, AdventHealth, a nonprofit organization that supports education and research. L.M.L. serves as a consultant for Boehringer Ingelheim, Eli Lilly, Novo Nordisk, Janssen, Dexcom, Insulet, Roche, Medtronic, Dompe, Provention Bio, and Vertex. CM serves, or has served, on the advisory panel for Novo Nordisk, Sanofi, Merck Sharp and Dohme Ltd., Eli Lilly and Company, Novartis, AstraZeneca, Boehringer Ingelheim, Roche, Medtronic, ActoBio Therapeutics, Pfizer, Imcyse, Insulet, Zealand Pharma, Avotres and Vertex. Financial compensation for these activities has been received by KU Leuven; KU Leuven has received research support for CM from Medtronic, Imcyse, Novo Nordisk, Sanofi, and ActoBio Therapeutics. L.M.J., E.C., A.C., S.M.P., M.H., S.C., M.U., S.A., M.T., M.J.R., and J.J.W. report no dualities of interest.

## Additional information

## ADA/EASD PMDI

Deirdre K. Tobias[11,12], Jordi Merino[13,14,15], Abrar Ahmad[16], Catherine Aiken[17,18], Jamie L. Benham[19], Dhanasekaran Bodhini[20], Amy L. Clark[21], Kevin Colclough[22], Rosa Corcoy[23,24,25], Sara J. Cromer[14,26,27], Daisy Duan[28], Jamie L. Felton[29,30,31], Ellen C. Francis[32], Pieter Gillard[3], Véronique Gingras[33,34], Romy Gaillard[35], Eram Haider[36], Alice Hughes[22], Jennifer M. Ikle[37,38], Laura M. Jacobsen[1,197], Anna R. Kahkoska[39], Jarno L. T. Kettunen[40,41,42], Raymond J. Kreienkamp[14,15,26,43], Lee-Ling Lim[44,45,46], Jonna M. E. Männistö[47,48], Robert Massey[36], Niamh-Maire Mclennan[49], Rachel G. Miller[50],

Mario Luca Morieri[51,52], Jasper Most[53], Rochelle N. Naylor[54], Bige Ozkan[55,56], Kashyap Amratlal Patel[22], Scott J. Pilla[57,58], Katsiaryna Prystupa[59,60], Sridaran Raghaven[61,62], Mary R. Rooney[55,63], Martin Schön[59,60,64], Zhila Semnani-Azad[12], Magdalena Sevilla-Gonzalez[26,27,65], Pernille Svalastoga[66,67], Wubet Worku Takele[68], Claudia Ha-ting Tam[46,69,70], Anne Cathrine B. Thuesen[13], Mustafa Tosur[5,71,72], Amelia S. Wallace[55,63], Caroline C. Wang[63], Jessie J. Wong[6], Jennifer M. Yamamoto[73], Katherine Young[22], Chloé Amouyal[74,75], Mette K. Andersen[13], Maxine P. Bonham[76], Mingling Chen[77], Feifei Cheng[78], Tinashe Chikowore[27,79,80,81], Sian C. Chivers[82], Christoffer Clemmensen[13], Dana Dabelea[83], Adem Y. Dawed[36], Aaron J. Deutsch[15,26,27], Laura T. Dickens[84], Linda A. DiMeglio[29,30,31,85], Monika Dudenhöffer-Pfeifer[16], Carmella Evans-Molina[29,30,31,86], María Mercè Fernández-Balsells[87,88], Hugo Fitipaldi[16], Stephanie L. Fitzpatrick[89], Stephen E. Gitelman[90], Mark O. Goodarzi[91,92], Jessica A. Grieger[93,94], Marta Guasch-Ferré[12,95], Nahal Habibi[93,94], Torben Hansen[13], Chuiguo Huang[46,69], Arianna Harris-Kawano[29,30,31], Heba M. Ismail[29,30,31], Benjamin Hoag[96,97], Randi K. Johnson[98,99], Angus G. Jones[22,100], Robert W. Koivula[101], Aaron Leong[14,27,102], Gloria K. W. Leung[76], Ingrid M. Libman[103], Kai Liu[93], S. Alice Long[104], William L. Lowe Jr.[105], Robert W. Morton[106,107,108], Ayesha A. Motala[109], Suna Onengut-Gumuscu[110], James S. Pankow[111], Maleesa Pathirana[93,94], Sofia Pazmino[112], Dianna Perez[29,30,31], John R. Petrie[113], Camille E. Powe[14,26,27,114], Alejandra Quinteros[93], Rashmi Jain[115,116], Debashree Ray[63,117], Mathias Ried-Larsen[118,119], Zeb Saeed[120], Vanessa Santhakumar[11], Sarah Kanbour[57,121], Sudipa Sarkar[57], Gabriela S. F. Monaco[29,30,31], Denise M. Scholtens[122], Elizabeth Selvin[55,63], Wayne Huey-Herng Sheu[123,124,125], Cate Speake[126], Maggie A. Stanislawski[98], Nele Steenackers[112], Andrea K. Steck[127], Norbert Stefan[60,128,129], Julie Støy[130], Rachael Taylor[131], Sok Cin Tye[132,133], Gebresilasea Gendisha Ukke[68], Marzhan Urazbayeva[72,134], Bart Van der Schueren[112,135], Camille Vatier[136,137], John M. Wentworth[138,139,140], Wesley Hannah[141,142], Sara L. White[82,143], Gechang Yu[46,69], Yingchai Zhang[46,69], Shao J. Zhou[94,144], Jacques Beltrand[145,146], Michel Polak[145,146], Ingvild Aukrust[66,147], Elisa de Franco[22], Sarah E. Flanagan[22], Kristin A. Maloney[148], Andrew McGovern[22], Janne Molnes[66,147], Mariam Nakabuye[13], Pål Rasmus Njølstad[66,67], Hugo Pomares-Millan[16,149], Michele Provenzano[150], Cécile Saint-Martin[151], Cuilin Zhang[152,153], Yeyi Zhu[154,155], Sungyoung Auh[156], Russell de Souza[107,157], Andrea J. Fawcett[158,159], Chandra Gruber[160], Eskedar Getie Mekonnen[161,162], Emily Mixter[163], Diana Sherifali[107,164], Robert H. Eckel[165], John J. Nolan[166,167], Louis H. Philipson[163], Rebecca J. Brown[156], Liana K. Billings[168,169], Kristen Boyle[83], Tina Costacou[50], John M. Dennis[22], Jose C. Florez[14,15,26,27], Anna L. Gloyn[37,38,170], Maria F. Gomez[16,171], Peter A. Gottlieb[127], Siri Atma W. Greeley[172], Kurt Griffin[116,173], Andrew T. Hattersley[22,100], Irl B. Hirsch[8], Marie-France Hivert[14,174,175], Korey K. Hood[6], Jami L. Josefson[158], Soo Heon Kwak[176], Lori M. Laffel[10,198], Siew S. Lim[68], Ruth J. F. Loos[13,177], Ronald C. W. Ma[46,69,70], Chantal Mathieu[3,198✉], Nestoras Mathioudakis[57], James B. Meigs[27,102,178], Shivani Misra[179,180], Viswanathan Mohan[181], Rinki Murphy[182,183,184], Richard Oram[22,100], Katharine R. Owen[101,185], Susan E. Ozanne[186], Ewan R. Pearson[36], Wei Perng[83], Toni I. Pollin[148,187], Rodica Pop-Busui[188], Richard E. Pratley[9], Leanne M. Redman[189], Maria J. Redondo[71,72], Rebecca M. Reynolds[49], Robert K. Semple[49,190], Jennifer L. Sherr[2,197], Emily K. Sims[29,30,31], Arianne Sweeting[191,192], Tiinamaija Tuomi[40,139,42], Miriam S. Udler[14,15,26,27], Kimberly K. Vesco[193], Tina Vilsbøll[194,195], Robert Wagner[59,60,196], Stephen S. Rich[110] & Paul W. Franks[12,16,101,108]

[11]Division of Preventative Medicine, Department of Medicine, Brigham and Women's Hospital and Harvard Medical School, Boston, MA, USA. [12]Department of Nutrition, Harvard T.H. Chan School of Public Health, Boston, MA, USA. [13]Novo Nordisk Foundation Center for Basic Metabolic Research, Faculty of Health and Medical Sciences, University of Copenhagen, Copenhagen, Denmark. [14]Diabetes Unit, Endocrine Division, Massachusetts General Hospital, Boston, MA, USA. [15]Center for Genomic Medicine, Massachusetts General Hospital, Boston, MA, USA. [16]Department of Clinical Sciences, Lund University Diabetes Centre, Lund University, Malmö, Sweden. [17]Department of Obstetrics and Gynaecology, the Rosie Hospital, Cambridge, UK. [18]NIHR Cambridge Biomedical Research Centre, University of Cambridge, Cambridge, UK. [19]Departments of Medicine and Community Health Sciences, Cumming School of Medicine, University of Calgary, Calgary, AB, Canada. [20]Department of Molecular Genetics, Madras Diabetes Research Foundation, Chennai, India. [21]Division of Pediatric Endocrinology, Department of Pediatrics, Saint Louis University School of Medicine, SSM Health Cardinal Glennon Children's Hospital, St. Louis, MO, USA. [22]Department of

Clinical and Biomedical Sciences, University of Exeter Medical School, Exeter, Devon, UK. [23]CIBER-BBN, ISCIII, Madrid, Spain. [24]Institut d'Investigació Biomèdica Sant Pau (IIB SANT PAU), Barcelona, Spain. [25]Departament de Medicina, Universitat Autònoma de Barcelona, Bellaterra, Spain. [26]Programs in Metabolism and Medical & Population Genetics, Broad Institute, Cambridge, MA, USA. [27]Department of Medicine, Harvard Medical School, Boston, MA, USA. [28]Division of Endocrinology, Diabetes and Metabolism, Johns Hopkins University School of Medicine, Baltimore, MD, USA. [29]Department of Pediatrics, Indiana University School of Medicine, Indianapolis, IN, USA. [30]Herman B Wells Center for Pediatric Research, Indiana University School of Medicine, Indianapolis, IN, USA. [31]Center for Diabetes and Metabolic Diseases, Indiana University School of Medicine, Indianapolis, IN, USA. [32]Department of Biostatistics and Epidemiology, Rutgers School of Public Health, Piscataway, NJ, USA. [33]Department of Nutrition, Université de Montréal, Montreal, Quebec, Canada. [34]Research Center, Sainte-Justine University Hospital Center, Montreal, Quebec, Canada. [35]Department of Pediatrics, Erasmus Medical Center, Rotterdam, The Netherlands. [36]Division of Population Health & Genomics, School of Medicine, University of Dundee, Dundee, UK. [37]Department of Pediatrics, Stanford School of Medicine, Stanford University, Stanford, CA, USA. [38]Stanford Diabetes Research Center, Stanford School of Medicine, Stanford University, Stanford, CA, USA. [39]Department of Nutrition, University of North Carolina at Chapel Hill, Chapel Hill, NC, USA. [40]Helsinki University Hospital, Abdominal Centre/Endocrinology, Helsinki, Finland. [41]Folkhalsan Research Center, Helsinki, Finland. [42]Institute for Molecular Medicine Finland FIMM, University of Helsinki, Helsinki, Finland. [43]Department of Pediatrics, Division of Endocrinology, Boston Children's Hospital, Boston, MA, USA. [44]Department of Medicine, Faculty of Medicine, University of Malaya, Kuala Lumpur, Malaysia. [45]Asia Diabetes Foundation, Hong Kong SAR, China. [46]Department of Medicine & Therapeutics, Chinese University of Hong Kong, Hong Kong SAR, China. [47]Departments of Pediatrics and Clinical Genetics, Kuopio University Hospital, Kuopio, Finland. [48]Department of Medicine, University of Eastern Finland, Kuopio, Finland. [49]Centre for Cardiovascular Science, Queen's Medical Research Institute, University of Edinburgh, Edinburgh, UK. [50]Department of Epidemiology, University of Pittsburgh, Pittsburgh, PA, USA. [51]Metabolic Disease Unit, University Hospital of Padova, Padova, Italy. [52]Department of Medicine, University of Padova, Padova, Italy. [53]Department of Orthopedics, Zuyderland Medical Center, Sittard-Geleen, The Netherlands. [54]Departments of Pediatrics and Medicine, University of Chicago, Chicago, IL, USA. [55]Welch Center for Prevention, Epidemiology, and Clinical Research, Johns Hopkins Bloomberg School of Public Health, Baltimore, MD, USA. [56]Ciccarone Center for the Prevention of Cardiovascular Disease, Johns Hopkins School of Medicine, Baltimore, MD, USA. [57]Department of Medicine, Johns Hopkins University, Baltimore, MD, USA. [58]Department of Health Policy and Management, Johns Hopkins University Bloomberg School of Public Health, Baltimore, MD, USA. [59]Institute for Clinical Diabetology, German Diabetes Center, Leibniz Center for Diabetes Research at Heinrich Heine University Düsseldorf, Auf'm Hennekamp 65, 40225 Düsseldorf, Germany. [60]German Center for Diabetes Research (DZD), Ingolstädter Landstraße 1, 85764 Neuherberg, Germany. [61]Section of Academic Primary Care, US Department of Veterans Affairs Eastern Colorado Health Care System, Aurora, CO, USA. [62]Department of Medicine, University of Colorado School of Medicine, Aurora, CO, USA. [63]Department of Epidemiology, Johns Hopkins Bloomberg School of Public Health, Baltimore, MD, USA. [64]Institute of Experimental Endocrinology, Biomedical Research Center, Slovak Academy of Sciences, Bratislava, Slovakia. [65]Clinical and Translational Epidemiology Unit, Massachusetts General Hospital, Boston, MA, USA. [66]Mohn Center for Diabetes Precision Medicine, Department of Clinical Science, University of Bergen, Bergen, Norway. [67]Children and Youth Clinic, Haukeland University Hospital, Bergen, Norway. [68]Eastern Health Clinical School, Monash University, Melbourne, VIC, Australia. [69]Laboratory for Molecular Epidemiology in Diabetes, Li Ka Shing Institute of Health Sciences, The Chinese University of Hong Kong, Hong Kong, China. [70]Hong Kong Institute of Diabetes and Obesity, The Chinese University of Hong Kong, Hong Kong, China. [71]Department of Pediatrics, Baylor College of Medicine, Houston, TX, USA. [72]Division of Pediatric Diabetes and Endocrinology, Texas Children's Hospital, Houston, TX, USA. [73]Internal Medicine, University of Manitoba, Winnipeg, MB, Canada. [74]Department of Diabetology, APHP, Paris, France. [75]Sorbonne Université, INSERM, NutriOmic team, Paris, France. [76]Department of Nutrition, Dietetics and Food, Monash University, Melbourne, VIC, Australia. [77]Monash Centre for Health Research and Implementation, Monash University, Clayton, VIC, Australia. [78]Health Management Center, The Second Affiliated Hospital of Chongqing Medical University, Chongqing Medical University, Chongqing, China. [79]MRC/Wits Developmental Pathways for Health Research Unit, Department of Paediatrics, Faculty of Health Sciences, University of the Witwatersrand, Johannesburg, South Africa. [80]Channing Division of Network Medicine, Brigham and Women's Hospital, Boston, MA, USA. [81]Sydney Brenner Institute for Molecular Bioscience, Faculty of Health Sciences, University of the Witwatersrand, Johannesburg, South Africa. [82]Department of Women and Children's health, King's College London, London, UK. [83]Lifecourse Epidemiology of Adiposity and Diabetes (LEAD) Center, University of Colorado Anschutz Medical Campus, Aurora, CO, USA. [84]Section of Adult and Pediatric Endocrinology, Diabetes and Metabolism, Kovler Diabetes Center, University of Chicago, Chicago, IL, USA. [85]Department of Pediatrics, Riley Hospital for Children, Indiana University School of Medicine, Indianapolis, IN, USA. [86]Richard L. Roudebush VAMC, Indianapolis, IN, USA. [87]Biomedical Research Institute Girona, IdIBGi, Girona, Spain. [88]Diabetes, Endocrinology and Nutrition Unit Girona, University Hospital Dr Josep Trueta, Girona, Spain. [89]Institute of Health System Science, Feinstein Institutes for Medical Research, Northwell Health, Manhasset, NY, USA. [90]University of California at San Francisco, Department of Pediatrics, Diabetes Center, San Francisco, CA, USA. [91]Division of Endocrinology, Diabetes and Metabolism, Cedars-Sinai Medical Center, Los Angeles, CA, USA. [92]Department of Medicine, Cedars-Sinai Medical Center, Los Angeles, CA, USA. [93]Adelaide Medical School, Faculty of Health and Medical Sciences, The University of Adelaide, Adelaide, Australia. [94]Robinson Research Institute, The University of Adelaide, Adelaide, Australia. [95]Department of Public Health and Novo Nordisk Foundation Center for Basic Metabolic Research, Faculty of Health and Medical Sciences, University of Copenhagen, 1014 Copenhagen, Denmark. [96]Division of Endocrinology and Diabetes, Department of Pediatrics, Sanford Children's Hospital, Sioux Falls, SD, USA. [97]University of South Dakota School of Medicine, E Clark St, Vermillion, SD, USA. [98]Department of Biomedical Informatics, University of Colorado Anschutz Medical Campus, Aurora, CO, USA. [99]Department of Epidemiology, Colorado School of Public Health, Aurora, CO, USA. [100]Royal Devon University Healthcare NHS Foundation Trust, Exeter, UK. [101]Oxford Centre for Diabetes, Endocrinology and Metabolism, University of Oxford, Oxford, UK. [102]Division of General Internal Medicine, Massachusetts General Hospital, Boston, MA, USA. [103]UPMC Children's Hospital of Pittsburgh, Pittsburgh, PA, USA. [104]Center for Translational Immunology, Benaroya Research Institute, Seattle, WA, USA. [105]Department of Medicine, Northwestern University Feinberg School of Medicine, Chicago, IL, USA. [106]Department of Pathology & Molecular Medicine, McMaster University, Hamilton, Canada. [107]Population Health Research Institute, Hamilton, Canada. [108]Department of Translational Medicine, Medical Science, Novo Nordisk Foundation, Tuborg Havnevej 19, 2900 Hellerup, Denmark. [109]Department of Diabetes and Endocrinology, Nelson R Mandela School of Medicine, University of KwaZulu-Natal, Durban, South Africa. [110]Center for Public Health Genomics, Department of Public Health Sciences, University of Virginia, Charlottesville, VA, USA. [111]Division of Epidemiology and Community Health, School of Public Health, University of Minnesota, Minneapolis, MN, USA. [112]Department of Chronic Diseases and Metabolism, Clinical and Experimental Endocrinology, KU Leuven, Leuven, Belgium. [113]School of Health and Wellbeing, College of Medical, Veterinary and Life Sciences, University of Glasgow, Glasgow, UK. [114]Department of Obstetrics, Gynecology, and Reproductive Biology, Massachusetts General Hospital and Harvard Medical School, Boston, MA, USA. [115]Sanford Children's Specialty Clinic, Sioux Falls, SD, USA. [116]Department of Pediatrics, Sanford School of Medicine, University of South Dakota, Sioux Falls, SD, USA. [117]Department of Biostatistics, Johns Hopkins Bloomberg School of Public Health, Baltimore, Maryland, USA. [118]Centre for Physical Activity Research, Rigshospitalet, Copenhagen, Denmark. [119]Institute for Sports and Clinical Biomechanics, University of Southern Denmark, Odense, Denmark. [120]Department of Medicine, Division of Endocrinology, Diabetes and

Metabolism, Indiana University School of Medicine, Indianapolis, IN, USA. [121]AMAN Hospital, Doha, Qatar. [122]Department of Preventive Medicine, Division of Biostatistics, Northwestern University Feinberg School of Medicine, Chicago, IL, USA. [123]Institute of Molecular and Genomic Medicine, National Health Research Institutes, Taipei, Taiwan. [124]Divsion of Endocrinology and Metabolism, Taichung Veterans General Hospital, Taichung, Taiwan. [125]Division of Endocrinology and Metabolism, Taipei Veterans General Hospital, Taipei, Taiwan. [126]Center for Interventional Immunology, Benaroya Research Institute, Seattle, WA, USA. [127]Barbara Davis Center for Diabetes, University of Colorado Anschutz Medical Campus, Aurora, CO, USA. [128]University Hospital of Tübingen, Tübingen, Germany. [129]Institute of Diabetes Research and Metabolic Diseases (IDM), Helmholtz Center Munich, Neuherberg, Germany. [130]Steno Diabetes Center Aarhus, Aarhus University Hospital, Aarhus, Denmark. [131]University of Newcastle, Newcastle upon Tyne, UK. [132]Sections on Genetics and Epidemiology, Joslin Diabetes Center, Harvard Medical School, Boston, MA, USA. [133]Department of Clinical Pharmacy and Pharmacology, University Medical Center Groningen, Groningen, The Netherlands. [134]Gastroenterology, Baylor College of Medicine, Houston, TX, USA. [135]Department of Endocrinology, University Hospitals Leuven, Leuven, Belgium. [136]Sorbonne University, Inserm U938, Saint-Antoine Research Centre, Institute of Cardiometabolism and Nutrition, Paris 75012, France. [137]Department of Endocrinology, Diabetology and Reproductive Endocrinology, Assistance Publique-Hôpitaux de Paris, Saint-Antoine University Hospital, National Reference Center for Rare Diseases of Insulin Secretion and Insulin Sensitivity (PRISIS), Paris, France. [138]Royal Melbourne Hospital Department of Diabetes and Endocrinology, Parkville, Vic, Australia. [139]Walter and Eliza Hall Institute, Parkville, VIC, Australia. [140]University of Melbourne Department of Medicine, Parkville, VIC, Australia. [141]Deakin University, Melbourne, Australia. [142]Department of Epidemiology, Madras Diabetes Research Foundation, Chennai, India. [143]Department of Diabetes and Endocrinology, Guy's and St Thomas' Hospitals NHS Foundation Trust, London, UK. [144]School of Agriculture, Food and Wine, University of Adelaide, Adelaide, Australia. [145]Institut Cochin, Inserm U, 10116 Paris, France. [146]Pediatric endocrinology and diabetes, Hopital Necker Enfants Malades, APHP Centre, université de Paris, Paris, France. [147]Department of Medical Genetics, Haukeland University Hospital, Bergen, Norway. [148]Department of Medicine, University of Maryland School of Medicine, Baltimore, MD, USA. [149]Department of Epidemiology, Geisel School of Medicine at Dartmouth, Hanover, NH, USA. [150]Nephrology, Dialysis and Renal Transplant Unit, IRCCS—Azienda Ospedaliero-Universitaria di Bologna, Alma Mater Studiorum University of Bologna, Bologna, Italy. [151]Department of Medical Genetics, AP-HP Pitié-Salpêtrière Hospital, Sorbonne University, Paris, France. [152]Global Center for Asian Women's Health, Yong Loo Lin School of Medicine, National University of Singapore, Queenstown, Singapore. [153]Department of Obstetrics and Gynecology, Yong Loo Lin School of Medicine, National University of Singapore, Queenstown, Singapore. [154]Kaiser Permanente Northern California Division of Research, Oakland, CA, USA. [155]Department of Epidemiology and Biostatistics, University of California, San Francisco, CA, USA. [156]National Institute of Diabetes and Digestive and Kidney Diseases, National Institutes of Health, Bethesda, MD, USA. [157]Department of Health Research Methods, Evidence, and Impact, Faculty of Health Sciences, McMaster University, Hamilton, ON, Canada. [158]Ann & Robert H. Lurie Children's Hospital of Chicago, Department of Pediatrics, Northwestern University Feinberg School of Medicine, Chicago, IL, USA. [159]Department of Clinical and Organizational Development, Chicago, IL, USA. [160]American Diabetes Association, Arlington, VA, USA. [161]College of Medicine and Health Sciences, University of Gondar, Gondar, Ethiopia. [162]Global Health Institute, Faculty of Medicine and Health Sciences, University of Antwerp, 2160 Antwerp, Belgium. [163]Department of Medicine and Kovler Diabetes Center, University of Chicago, Chicago, IL, USA. [164]School of Nursing, Faculty of Health Sciences, McMaster University, Hamilton, Canada. [165]Division of Endocrinology, Metabolism, Diabetes, University of Colorado, Aurora, CO, USA. [166]Department of Clinical Medicine, School of Medicine, Trinity College Dublin, Dublin, Ireland. [167]Department of Endocrinology, Wexford General Hospital, Wexford, Ireland. [168]Division of Endocrinology, NorthShore University HealthSystem, Skokie, IL, USA. [169]Department of Medicine, Prtizker School of Medicine, University of Chicago, Chicago, IL, USA. [170]Department of Genetics, Stanford School of Medicine, Stanford University, CA, USA. [171]Faculty of Health, Aarhus University, Aarhus, Denmark. [172]Departments of Pediatrics and Medicine and Kovler Diabetes Center, University of Chicago, Chicago, USA. [173]Sanford Research, Sioux Falls, SD, USA. [174]Department of Population Medicine, Harvard Medical School, Harvard Pilgrim Health Care Institute, Boston, MA, USA. [175]Department of Medicine, Universite de Sherbrooke, Sherbrooke, QC, Canada. [176]Department of Internal Medicine, Seoul National University College of Medicine, Seoul National University Hospital, Seoul, Republic of Korea. [177]Charles Bronfman Institute for Personalized Medicine, Icahn School of Medicine at Mount Sinai, New York, NY, USA. [178]Broad Institute, Cambridge, MA, USA. [179]Division of Metabolism, Digestion and Reproduction, Imperial College London, London, UK. [180]Department of Diabetes & Endocrinology, Imperial College Healthcare NHS Trust, London, UK. [181]Department of Diabetology, Madras Diabetes Research Foundation & Dr. Mohan's Diabetes Specialities Centre, Chennai, India. [182]Department of Medicine, Faculty of Medicine and Health Sciences, University of Auckland, Auckland, New Zealand. [183]Auckland Diabetes Centre, Te Whatu Ora Health New Zealand, Auckland, New Zealand. [184]Medical Bariatric Service, Te Whatu Ora Counties, Health New Zealand, Auckland, New Zealand. [185]Oxford NIHR Biomedical Research Centre, University of Oxford, Oxford, UK. [186]University of Cambridge, Metabolic Research Laboratories and MRC Metabolic Diseases Unit, Wellcome-MRC Institute of Metabolic Science, Cambridge, UK. [187]Department of Epidemiology & Public Health, University of Maryland School of Medicine, Baltimore, MD, USA. [188]Department of Internal Medicine, Division of Metabolism, Endocrinology and Diabetes, University of Michigan, Ann Arbor, MI, USA. [189]Pennington Biomedical Research Center, Baton Rouge, LA, USA. [190]MRC Human Genetics Unit, Institute of Genetics and Cancer, University of Edinburgh, Edinburgh, UK. [191]Faculty of Medicine and Health, University of Sydney, Sydney, NSW, Australia. [192]Department of Endocrinology, Royal Prince Alfred Hospital, Sydney, NSW, Australia. [193]Kaiser Permanente Northwest, Kaiser Permanente Center for Health Research, Portland, OR, USA. [194]Clinial Research, Steno Diabetes Center Copenhagen, Herlev, Denmark. [195]Department of Clinical Medicine, Faculty of Health and Medical Sciences, University of Copenhagen, Copenhagen, Denmark. [196]Department of Endocrinology and Diabetology, University Hospital Düsseldorf, Heinrich Heine University Düsseldorf, Moorenstr. 5, 40225 Düsseldorf, Germany.

