## [Peer Review File · Communications Medicine]

Reviewers' comments:

Reviewer #1 (Remarks to the Author):

This is a well-written and comprehensive review article on an important topic. It covers relevant details of the literature and adds novelty to the subject of diabetes technologies.

There are a few aspects which merit further attention:

1. Can the authors advise the involvement of industry in the trials reported? i.e. how many of the articles reported had direct industry involvement? Can the authors comment on this in their discussion if this is a limitation?
2. I am pleased that the authors comment on low levels of ethnic and lower socio-economic deprivation. This is relevant towards African-Afro-Caribbean and Asian populations, in addition to the non-white hispanic populations mentioned. Perhaps this could be included in the relevant area in the manuscript. Similarly have pwT1D with high levels of complications or high HbA1c adequately been included in trials?
3. Although numerical benefits to TIR/A1c were highlighted, realising the limitations mentioned in PROs, have psychological benefits in T1D been adequately addressed by the current trials studied?
4. Have the trials adequately reported real-world issues eg skin adhesive issues/reactions, errors, device burden/distress or technology fatigue?
5. I am pleased that the authors comment on the use of real-world evidence. Given the pace at which technology is moving, issues with adequate population inclusion in RCTs and head to head trials, as well as opportunities for pwT1D to adapt towards open-source approaches, this aspects may benefit from a further comment in the discussion with reference to current registry / audit/ real-world data that are already being reported and comment on whether these support findings from RCTs.

Reviewer #2 (Remarks to the Author):

This is an excellent systematic review of the evidence base for novel technologies (glucose monitoring, insulin pumps and decision support) in people with type 1 diabetes by a stellar group which will have a high impact in the healthcare professional community. I cannot find fault in the technical implementation of the review and have very few criticisms. The information provided while not novel is logically organised as part of a single comprehensive document.

Major Points

1/ The stated "The primary goal of the review was to determine whether characteristics of people living with type 1 diabetes across the lifespan could identify the best (tailored) treatment(s) to optimize outcomes, including HbA1c, TIR, weight, hypoglycemia, quality of life, and other measures." The manuscript does also devote significant space to the overall impact of these technologies before the precision assessment is detailed and this is helpful as it contextualises the subsequent precision analysis.

It would however be worthwhile formally defining these precision characteristics a priori in the methods section (or were these identified post hoc based on what was available in the publications?). The precision characteristics are first detailed in the Results section of the manuscript under the Precision Assessment Heading (age, metabolic status, race and ethnicity, nephropathy, and impaired awareness of hypoglycaemia). Are there others that need to be evaluated in the eyes

of the authors where there are no data available? eg end stage renal failure, psychiatric disorders including anorexia nervosa, people who fast for cultural reasons...

2/ In the tables could the authors please consider including a column describing the system used eg Tandem / Dexcom G5, Medtronic 670G/ Guardian Sensor etc. These devices / systems are not identical and PROs particularly may be influenced by the system's characteristics. Related to this, it would be worthwhile discussing technology design and its potential role in precision medicine in the context of PROs (for example a patch pump vs. a tubed device) and comorbidities such as visual impairment, reduced manual dexterity and cognition which would have impacted participant inclusion.

3/ Under decision support, have there been no studies on the benefit of software platforms such as Carelink, Clarity, Diasend/ Glooko?

4/ Broad statements such as "all cohorts of adults, even stratified within ages 18 to 87 years, had-significant metabolic benefits from CGM, LGS/PLGS and AID systems," require qualification in the discussion based upon an understanding of the limitations of the study design and technology. For example, older people recruited for these studies generally represent a highly functioning subset with intact cognition and fewer comorbidities in comparison with the mainstream older population.

Minor Points

1/ while they have justified the restriction to publications within the last 10 years could the authors please define how they came to decide upon including only those trials with ≥ 50 participants. Did they consider trials with smaller numbers to be inadequately powered?

2/ While it may go without saying, it should be made clear that this review related to the use of these technologies in the ambulatory setting and excluded hospitalised patients.

3/ Statements such as "Despite the rigorous procedures used, some AID trials that met inclusion criteria were not identified." Could the authors please provide brief details as to how these publications were missed by the search and subsequently unearthed?

We would like to thank the reviewers for their time and review of our manuscript, which we believe has strengthened our paper. Below in bold please find responses to points raised as well as details regarding where the manuscript has been updated accordingly.

Reviewers' comments:

Reviewer #1 (Remarks to the Author):

This is a well-written and comprehensive review article on an important topic. It covers relevant details of the literature and adds novelty to the subject of diabetes technologies.

There are a few aspects which merit further attention:

1. Can the authors advise the involvement of industry in the trials reported? i.e. how many of the articles reported had direct industry involvement? Can the authors comment on this in their discussion if this is a limitation?

Thank you for this suggestion. This information was extracted from the articles and has now been added into table 1. Additionally, in the relevant sections we have indicated percentages that were supported by industry.

A line has been added to the discussion on page 24 regarding potential limitations secondary to industry sponsorship of some of the studies. See below:

Not surprisingly, but important to note, device manufacturers sponsored nearly half (44%) of the of the trials conducted, with in-kind support of devices noted in another 18% of the trials identified.

2. I am pleased that the authors comment on low levels of ethnic and lower socio-economic deprivation. This is relevant towards African-Afro-Caribbean and Asian populations, in addition to the non-white hispanic populations mentioned. Perhaps this could be included in the relevant area in the manuscript. Similarly have pwT1D with high levels of complications or high HbA1c adequately been included in trials?

Thank you for this comment. We have added a line to the discussion on page 24 regarding limited inclusion of those with suboptimal glycemia and those with complications from diabetes in the present trials.

3. Although numerical benefits to TIR/A1c were highlighted, realising the limitations mentioned in PROs, have psychological benefits in T1D been adequately addressed by the current trials studied?

We appreciate the reviewer's comments. There may be limitations in PROs but consideration of psychological benefits of technology is important. However, current PROs do not adequately address cognitive burden and measures of sleep are not always included in the studies. We have added the following to the discussion on the bottom of page 25 to recognize this:

Recognizing that treatment success is not merely confined to glycemic metrics, PROs are now gaining traction in clinical trials. While these tools help to better describe potential benefits, or drawbacks, to various technologies, even they have limitations, like

the inability to fully assess cognitive burden. Greater psychological benefit may not be identified unless mixed method approaches are utilized, highlighting the need for more research in this area. Qualitative analysis will increase both the breadth and depth of data collected, which may allow greater focus on device burden and diabetes distress experienced by people with diabetes.

4. Have the trials adequately reported real-world issues eg skin adhesive issues/reactions, errors, device burden/distress or technology fatigue?

In our present systematic review, which was based on RCTs, these important areas were outside the focus of the data we collected. Yet, we agree with the reviewer that assessment of such issues is vastly important, and would likely best be assessed in real-world studies given the application of technology in a broader population. The discussion has been amended, starting on the top of page 25, accordingly to highlight the need to explore the frequency of device malfunctions/errors as well as collect data on skin reactions that may be secondary to exposure to adhesives. Finally, the need for mixed methods approaches will be required to more completely characterize distress, technology fatigue, and cognitive burden. Please see above regarding how this was delineated in the discussion.

5. I am pleased that the authors comment on the use of real-world evidence. Given the pace at which technology is moving, issues with adequate population inclusion in RCTs and head to head trials, as well as opportunities for pwt1d to adapt towards open-source approaches, this aspects may benefit from a further comment in the discussion with reference to current registry / audit/ real-world data that are already being reported and comment on whether these support findings from RCTs.

Thank you for this comment. We have added more to the discussion at the top of page 25 regarding real-world and registry data.

Reviewer #2 (Remarks to the Author):

This is an excellent systematic review of the evidence base for novel technologies (glucose monitoring, insulin pumps and decision support) in people with type 1 diabetes by a stellar group which will have a high impact in the healthcare professional community. I cannot find fault in the technical implementation of the review and have very few criticisms. The information provided while not novel is logically organised as part of a single comprehensive document.

Major Points

1/ The stated “The primary goal of the review was to determine whether characteristics of people living with type 1 diabetes across the lifespan could identify the best (tailored) treatment(s) to optimize outcomes, including HbA1c, TIR, weight, hypoglycemia, quality of life, and other measures.”

The manuscript does also devote significant space to the overall impact of these technologies before the precision assessment is detailed and this is helpful as it contextualises the subsequent precision analysis.

It would however be worthwhile formally defining these precision characteristics a priori in the methods section (or were these identified post hoc based on what was available in the publications?). The precision characteristics are first detailed in the Results section of the manuscript under the Precision Assessment Heading (age, metabolic status, race and ethnicity, nephropathy, and impaired awareness of hypoglycaemia). Are there others that need to be evaluated in the eyes of the authors where there are no data available? eg end stage renal failure, psychiatric disorders including anorexia nervosa, people who fast for cultural reasons...

Thank you for your comment. In our PROSPERO protocol, which was published prior to our initial search, we identified a priori proposed subgroups that we hoped to analyze. These included the following list:

- Age (including pediatric and geriatric)
- Sex
- Race/ethnicity
- HbA1c
- C-peptide
- Hypoglycemia unawareness
- Body mass index
- Duration of diabetes
- Age at onset
- Complications of diabetes (including end-stage renal disease, vision loss, cardiovascular disease, neuropathy)
- Co-existing significant health problems (including cancer/terminal disease)
- Co-existing auto-immune diseases (including thyroid, coeliac disease)
- Cognitive impairment
- Social determinants of health (including insurance coverage, socioeconomic status)
- Education (including health literacy/numeracy)
- Family structure/support systems

The data provided in the precision analysis is based on what we could find when we extracted information from the identified trials that met our inclusion and exclusion criteria. As noted in this larger list many variables were considered, including end stage renal failure, but no publications were identified which described such subgroups.

The list of terms considered for subgroup analysis is now listed in the new Supplemental Table 4.

2/ In the tables could the authors please consider including a column describing the system used eg Tandem / Dexcom G5, Medtronic 670G/ Guardian Sensor etc. These devices / systems are not identical and PROs particularly may be influenced by the system's characteristics. Related to this, it would be worthwhile discussing technology design and its potential role in precision medicine in the context of PROs (for example a patch pump vs. a tubed device) and comorbidities such as visual impairment, reduced manual dexterity and cognition which would have impacted participant inclusion.

When the JDRF CGM trial was designed, while there were 3 types of CGM devices utilized, the outcome of interest was a comparison of sensor glucose monitoring as compared to blood glucose monitoring in regards to both glycemic outcomes and PROs. The goal was to compare methods of glucose monitoring, not individual devices.

In the present analysis, with no studies identified that performed direct head-to-head comparisons of devices and with differences in study design, duration, and outcomes measured, we have opted not to include device manufacturers and instead focus on how various categories of technologies impact the outcomes of interest.

Further, while exploration of what devices may best suite individual populations, including those with visual impairment or reduced dexterity or cognition, our search of randomized controlled trials with a minimum of 50 participants did not identify such subgroups. The following has been added to the discussion to highlight this point:

Indeed, examination of particular subgroups, including those with visual impairment or reduced manual dexterity or cognition, may help guide how best to pair glucose monitoring and insulin delivery modality to the needs of an individual. However, such trials are unlikely to be conducted as head-to-head randomized control trials, again highlighting the need for real-world data collection or use of registry data that would include more diverse populations allowing for exploration of a variety of user characteristics.

3/ Under decision support, have there been no studies on the benefit of software platforms such as Carelink, Clarity, Diasend/ Glooko?

In the present literature search, we did not include search terms for “software platforms.” While software platforms help clinicians view data they are not technologies that impact care delivery without interpretation by a clinician and therefore we deemed this outside of the focus of the present review. However, future research and examination of software platforms is absolutely warranted.

4/ Broad statements such as “all cohorts of adults, even stratified within ages 18 to 87 years, had- significant metabolic benefits from CGM, LGS/PLGS and AID systems,” require qualification in the discussion based upon an understanding of the limitations of the study design and technology. For example, older people recruited for these studies generally represent a highly functioning subset with intact cognition and fewer comorbidities in comparison with the mainstream older population.

Thank you for this comment. We agree that this is an important clarification. We have amended this sentence on page 20 to read :

For example, *adults enrolled in these clinical trials*, even stratified within ages 18 to 87 years, had significant metabolic benefits from CGM, SAP, LGS/PLGS, and AID systems.

Additionally, we have added this line regarding cognitive state and the needed for subpopulations studies to page 25:

Indeed, examination of particular subgroups, including those with visual impairment or reduced manual dexterity or cognition, may help guide how best to pair glucose monitoring and insulin delivery modality to the needs of an individual. However, such trials are unlikely to be conducted as head-to-head randomized control trials, again highlighting the need for real-world data collection or use of registry data that would

include more diverse populations allowing for exploration of a variety of user characteristics.

Minor Points

1/ while they have justified the restriction to publications within the last 10 years could the authors please define how they came to decide upon including only those trials with ≥ 50 participants. Did they consider trials with smaller numbers to be inadequately powered?

Early on when considering inclusion criteria for the trials, our group came to consensus that it was reasonable to limit sample size of studies that were included. This was based on concern that smaller trials may be underpowered and therefore be prone to type I or type II error.

The following has been added to the methods:

Recognizing that smaller studies may be prone to either type I or type II error, the group came to consensus regarding the need for a minimum sample size of 50 participants in order to include a trial.

2/ While it may go without saying, it should be made clear that this review related to the use of these technologies in the ambulatory setting and excluded hospitalised patients.

The following has been added to the methods:

Studies were only included if they occurred in ambulatory settings.

3/ Statements such as “Despite the rigorous procedures used, some AID trials that met inclusion criteria were not identified.” Could the authors please provide brief details as to how these publications were missed by the search and subsequently unearthed?

With the expertise of the group, it was evident on review of the final list of articles to be extracted, certain key articles had errantly been missed during the literature search. The technical cause of the program’s failing to capture such articles is unclear, and instead it is evident that the tools utilized are fallible. Therefore, with consensus, it was determined that inclusion of certain articles was warranted.

We have added additional clarification to the Discussion on page 22:

We still uncovered some newer as well as *unexplained omissions* of publications germane to our topic, highlighting the dynamic nature of treatment approaches to type 1 diabetes and *challenges to systematic review of heterogenous trials and technologies.*

Reviewers' comments:

Reviewer #1 (Remarks to the Author):

Thank you for the revised versions- I am happy with the responses provided and changes made in the updated manuscript.

Reviewer #2 (Remarks to the Author):

I thank the authors for their detailed responses which have largely addressed the points raised. However, while some of this information may be inferred from the sponsored studies, I would strongly recommend that a column be included in Table 1 identifying the CGM/ SAP/ AID system being studied as this is fundamental. The assumption is that all devices are equal which they may or may not be. The request was made not so much to imply that a head-to-head comparison should be made between manufacturers, but to provide the reader / clinician with greater understanding of the implications particularly with technology (devices that most of the readers will be familiar with) evolving as rapidly as it is even over a period as short as 10 years. For example, when looking at data that they have so diligently collected, one would like to be able to differentiate evidence from a study with a prototype or a first to market device (e.g. Medtronic 670G) vs. a more evolved system. In addition, different CGM systems do not provide results that are necessarily equivalent.

We would like to thank the reviewers for their time and review of our manuscript, which we believe has strengthened our paper. Below in bold please find responses to points raised as well as details regarding where the manuscript has been updated accordingly.

Reviewers' comments:

Reviewer #1 (Remarks to the Author):

Thank you for the revised versions- I am happy with the responses provided and changes made in the updated manuscript.

Reviewer #2 (Remarks to the Author):

I thank the authors for their detailed responses which have largely addressed the points raised. However, while some of this information may be inferred from the sponsored studies, I would strongly recommend that a column be included in Table 1 identifying the CGM/ SAP/ AID system being studied as this is fundamental. The assumption is that all devices are equal which they may or may not be. The request was made not so much to imply that a head-to-head comparison should be made between manufacturers, but to provide the reader / clinician with greater understanding of the implications particularly with technology (devices that most of the readers will be familiar with) evolving as rapidly as it is even over a period as short as 10 years. For example, when looking at data that they have so diligently collected, one would like to be able to differentiate evidence from a study with a prototype or a first to market device (e.g. Medtronic 670G) vs. a more evolved system. In addition, different CGM systems do not provide results that are necessarily equivalent.

Thank you for this comment. We agree and have added this information to Table 1.

REVIEWERS' COMMENTS:

Reviewer #2 (Remarks to the Author):

I thank the authors. They have responded to my satisfaction